# ANKS1A regulates LDL receptor-related protein 1 (LRP1)-mediated cerebrovascular clearance in brain endothelial cells

Jiyeon Lee[1,5], Haeryung Lee [1,5], Hyein Lee[2], Miram Shin[1], Min-Gi Shin [3], Jinsoo Seo [2], Eun Jeong Lee [3], Sun Ah Park [4] & Soochul Park [1]✉

Brain endothelial LDL receptor-related protein 1 (LRP1) is involved in the clearance of Aβ peptides across the blood-brain barrier (BBB). Here we show that endothelial deficiency of ankyrin repeat and SAM domain containing 1 A (ANKS1A) reduces both the cell surface levels of LRP1 and the Aβ clearance across the BBB. Association of ANKS1A with the NPXY motifs of LRP1 facilitates the transport of LRP1 from the endoplasmic reticulum toward the cell surface. ANKS1A deficiency in an Alzheimer's disease mouse model results in exacerbated Aβ pathology followed by cognitive impairments. These deficits are reversible by gene therapy with brain endothelial-specific *ANKS1A*. In addition, human induced pluripotent stem cell-derived BBBs (iBBBs) were generated from endothelial cells lacking *ANKS1A* or carrying the *rs6930932* variant. Those iBBBs exhibit both reduced cell surface LRP1 and impaired Aβ clearance. Thus, our findings demonstrate that ANKS1A regulates LRP1-mediated Aβ clearance across the BBB.

Production of amyloid beta (Aβ) peptides in neurons and their senile amyloid beta plaque formation is a prominent hallmark of Alzheimer's disease (AD)[1], and it is known that inefficient clearance of toxic Aβ peptides from the brain contributes to initiation and progression of AD[2,3]. Whereas astrocytes, microglia and neurons take up the toxic Aβ peptides for clearance[4–6], the vascular clearance of Aβ peptides represents a major route to override the build-up of Aβ peptides[2,7]. Pathological amyloid deposition in the walls of a cerebral vessel is termed cerebral amyloid angiopathy (CAA) with Aβ type-CAA having been observed as a neuropathological symptom in 48% of patients with AD[8]. It is known that CAA progressively disturbs the blood brain barrier (BBB) function, and renders affected individuals more vulnerable to cerebral bleeding and subsequently neuronal loss and cognitive decline[9–12].

Low-density lipoprotein receptor-related protein 1 (LRP1) is the most studied Aβ receptor[13], although it was first identified as an apolipoprotein E (APOE) receptor[14]. Functionally, endothelial-specific LRP1

mediates transcytosis and clearance of toxic Aβ peptides across the BBB[5,15–21], and evidence indicates that Aβ peptides bind to the LRP1 receptor on the abluminal surface of endothelial cells and the formed complexes are then internalized by phosphatidylinositol-binding clathrin assembly protein, PICALM, and the clathrin-mediated pathway[16,18,22–27]. The internalized complexes are routed to the transcytotic pathway by RAB5 and RAB11, and released into the circulating bloodstream[15,17–21,23,28,29]. For an aging brain, the overall levels of endothelial LRP1 decrease and, in the cerebral vessels of AD patients[16,23,30,31] and AD animal models[16,18,19,32], LRP1 levels are nearly not present. Reduced endothelial LRP1 levels correlate highly with BBB breakdown, followed by cognitive dysfunction[12,33–35]. Endothelial-specific LRP1 knockout (KO) is shown to activate the cyclophilin A (CypA)-matrix metalloproteinase-9 (MMP-9) pathway as a potential mechanism underlying progressive BBB impairment and neurodegeneration[15,36–38]. Therefore, maintaining the endothelial LRP1 levels may protect against BBB breakdown and neurodegeneration. For Aβ clearance from the

[1]Department of Biological Sciences, Sookmyung Women's University, Seoul 04310, Korea. [2]Department of Brain Sciences, Daegu Gyeongbuk Institute of Science & Technology (DGIST), Daegu 42988, Korea. [3]Department of Brain Science, Ajou University School of Medicine, Suwon 16499, Korea. [4]Lab for Neurodegenerative Dementia, Department of Anatomy, and Department of Neurology, Ajou University School of Medicine, Suwon 16499, Korea. [5]These authors contributed equally: Jiyeon Lee, Haeryung Lee. ✉e-mail: scpark@sookmyung.ac.kr

brain into the circulation, it is, however, unclear on how the homeostatic levels of LRP1 are maintained in the basolateral surface of endothelial cells.

LRP1 is an endocytic, signaling transmembrane protein, regulating diverse cellular functions in the brain and it binds to different cell signaling adaptors via its short cytoplasmic tail[18,39–42]. For internalization of LRP1, the cytoplasmic tail of LRP1 contains one YXXL motif, which is critical for this function[43,44]. PICALM binding to LRP1 also requires the YXXL motif and this interaction is crucial for the PICALM/clathrin-dependent internalization of endothelial LRP1[18,43,45–48]. LRP1 also includes two NPXY motifs in its cytoplasmic tail and multiple studies have shown a functional effect for these NPXY motifs[38,49–58]. In general, these motifs bind to phospho-tyrosine binding (PTB) domains of various proteins. For LRP1, PTB containing FE65[49] and Disabled-1 (DAB1)[50] bind to LRP1 NPXY motifs, and these have been implicated in amyloidogenic APP processing and subsequent Aβ production. Another PTB-containing binding partner for LRP1 is the ankyrin repeat and SAM domain-containing protein 1B (ANKS1B), which is a paralogue of ANKS1A[59–61].

It is noteworthy that the ANKS1 family of proteins have been implicated in regulating the endoplasmic reticulum (ER) export of certain membrane proteins including EphA2[62]. It remains unknown whether the ER export of LRP1 is regulated by the ANKS family of proteins or other PTB-containing adaptors. In this study, we observed that the ANKS1 family member, ANKS1A, is specifically expressed in the endothelial cells of the brain, similar to LRP1; furthermore, the changes in the expression levels of ANKS1A and LRP1 are regulated by the integrity of an intact BBB and the level of brain aging, respectively. We then hypothesized that ANKS1A plays a direct and central role in regulating the levels of LRP1, a key receptor for Aβ clearance for the brain endothelium. It was then shown that ANKS1A has a role in regulating LRP1-mediated Aβ clearance in various BBB models, including the model generated by human induced pluripotent stem cells (iPSCs); the effect of brain endothelial-specific ANKS1A gene therapy in these models also supported the role of ANKS1A in the LRP1 function.

## Results

### Regulation of the brain endothelial surface LRP1 by ANKS1A in the bEnd.3 cellular barrier model of BBB

LRP1 interacts with the ANKS1 family proteins[61], but a causative relationship in biological systems remains unknown. As shown in a recent report[63], we observed an age-related decrease in the expression of genes for ligand-specific receptor-mediated transcytosis (RMT) in brain microvessels (Supplementary Fig. 1a). Importantly, ANKS1A gene expression significantly decreased in microvessels of aged-brains, as did LRP1 and its downstream adaptor, PICALM (Supplementary Fig. 1a). When lipopolysaccharide (LPS) was used to disrupt the BBB in adult mice, LRP1 and other BBB-related gene expression levels significantly decreased and coincided with the reduced ANKS1A expression (Supplementary Fig. 1b); by comparison, the expression of the other ANKS1 member, ANKS1B, was barely detectable in the brain microvessels (lane 13). Using the reporter mice ANKS1A[+/LacZ] and ANKS1B[+/LacZ], we confirmed the expression of ANKS1A, but not ANKS1B, in the cerebrovascular tissue of the whole brain (Supplementary Fig. 1c), including the neocortex and the hippocampus (Fig. 1a). Immunohistochemical staining also suggested that ANKS1A is expressed in the brain endothelial cells rather than the pericytes (Supplementary Fig. 1d). However, ANKS1A expression based on X-gal staining was not detectable in astrocytes, microglia or oligodendrocytes (Supplementary Fig. 1e). We also analyzed the expression of ANKS1A in various cell types of the hippocampus from the published snRNAseq dataset (GSE 166261)[64] and uncovered ANKS1A highest expression being in brain endothelial cells (Supplementary Fig. 1f). To further confirm the brain endothelial specific expression of ANKS1A, the purified cerebral microvessels (Supplementary Fig. 1g) were dissociated into single cells

and the principal cells constituting the BBB were cultured for RT-qPCR analysis of various markers. In this system, pericytes, astrocytes and endothelial cells expressed the known cell type-specific markers (Supplementary Fig. 1h), and the levels of expression for ANKS1A were notable in dissociated endothelial cells (Supplementary Fig. 1h). However, ANKS1A expression seemed to be lower in cultured, dissociated endothelial cells than in undissociated brain microvessels (Supplementary Fig. 1a, b, h). This suggested that the endothelial expression of ANKS1A is regulated by its proximal interaction with pericytes and/or astrocytes. To address this issue, we used bEnd.3[65,66], an immortalized mouse brain endothelial cell line, to set up a co-culture system of various cell types, in which the bEnd.3 cells were supplied with soluble factors secreted from a mixture of astrocytes and pericytes in a transwell format (Supplementary Fig. 1i). RT-qPCR analysis of bEnd.3 cells showed an up-regulation of ANKS1A together with LRP1 and other BBB-specific markers in response to the soluble factors from the transwell setup (Supplementary Fig. 1j). Western blot analysis showed that the proteins levels of ANKS1A and other BBB-specific markers in bEnd.3 cells were also significantly elevated in the transwell co-culture with astrocytes and pericytes (Supplementary Fig. 1k). Taken together, our findings suggest that ANKS1A is specifically expressed in the brain endothelial cells and that its expression is promoted by an intact BBB.

We next tested whether the surface levels of LRP1 were dependent on ANKS1A levels in bEnd.3 cells using a specific anti-LRP1 antibody (see Supplementary Fig. 6n, o). Strikingly, in cells transfected with ANKS1A-specific siRNAs, but not scrambled siRNAs, the cell surface membrane staining of LRP1 was significantly reduced (Fig. 1b, c and Supplementary Fig. 1l). The polarized localization of LRP1 to the basolateral side of endothelial cells was not routed to their apical side and it was decreased by the reduced levels of ANKS1A in cells treated with ANKS1A-specific siRNAs (Supplementary Fig. 1m). In addition, as HyLite488-labeled Aβ(1–40) peptides bind more effectively to the surface of control cells at 4 °C and they are readily internalized at 37 °C, this effect was significantly reduced in cells transfected with ANKS1A siRNAs (Supplementary Fig. 1n). This finding led us to investigate whether ANKS1A regulates the LRP1-mediated transendothelial clearance of Aβ peptides. For this purpose, bEnd.3 cells were first cultured in the transwell insert; then, they were overlaid by astrocytes and pericytes, and this co-culture was further incubated for 12 days to form and serve as an in vitro model of the BBB (Fig. 1d). In this setting, the top of the transwell insert is the abluminal side whereas the bottom is the luminal side. Transendothelial electrical resistance (TEER) measurement[67] showed that bEnd.3 cells co-cultured with astrocytes and pericytes form the best in vitro model of the BBB, establishing a more effective barrier than the co-culture of bEnd.3 cells with mouse embryonic fibroblasts (MEFs); by comparison, MEFs with astrocytes and pericytes formed the least effective barrier in terms of their TEER values (Fig. 1e). In addition, the paracellular, bottom-to-top, permeability of either the 70 kDa dextran or the 4 kDa dextran was least for the in vitro BBB model, validating the effectiveness of this barrier model (Fig. 1f, g). We next asked whether the Aβ(1–40) peptides from the abluminal side were able to penetrate the above cellular barriers. Importantly, the in vitro BBB model with bEnd.3, astrocytes and pericytes allowed more Aβ(1–40) peptides to pass through the insert (Fig. 1h) and this permeability was effectively inhibited by the knockdown of RAB11A and RAB11B, the small GTPase paralogs critical for transcytosis[18] (Fig. 1i and Supplementary Fig. 1o). The in vitro BBB derived from ANKS1A siRNA-transfected bEnd.3 cells did not affect the paracellular permeability of either the 4 kDa dextran (Fig. 1j) or the Aβ(1–40) peptides (Fig. 1k). However, compared to the control in vitro BBB, the ANKS1A-deficient BBB was not as effective in facilitating the transcytosis of Aβ(1–40) peptides (Fig. 1m), while it did not affect the permeability of the 4 kDa dextran (Fig. 1l). Taken together, our results strongly suggest that ANKS1A regulates the surface levels of the

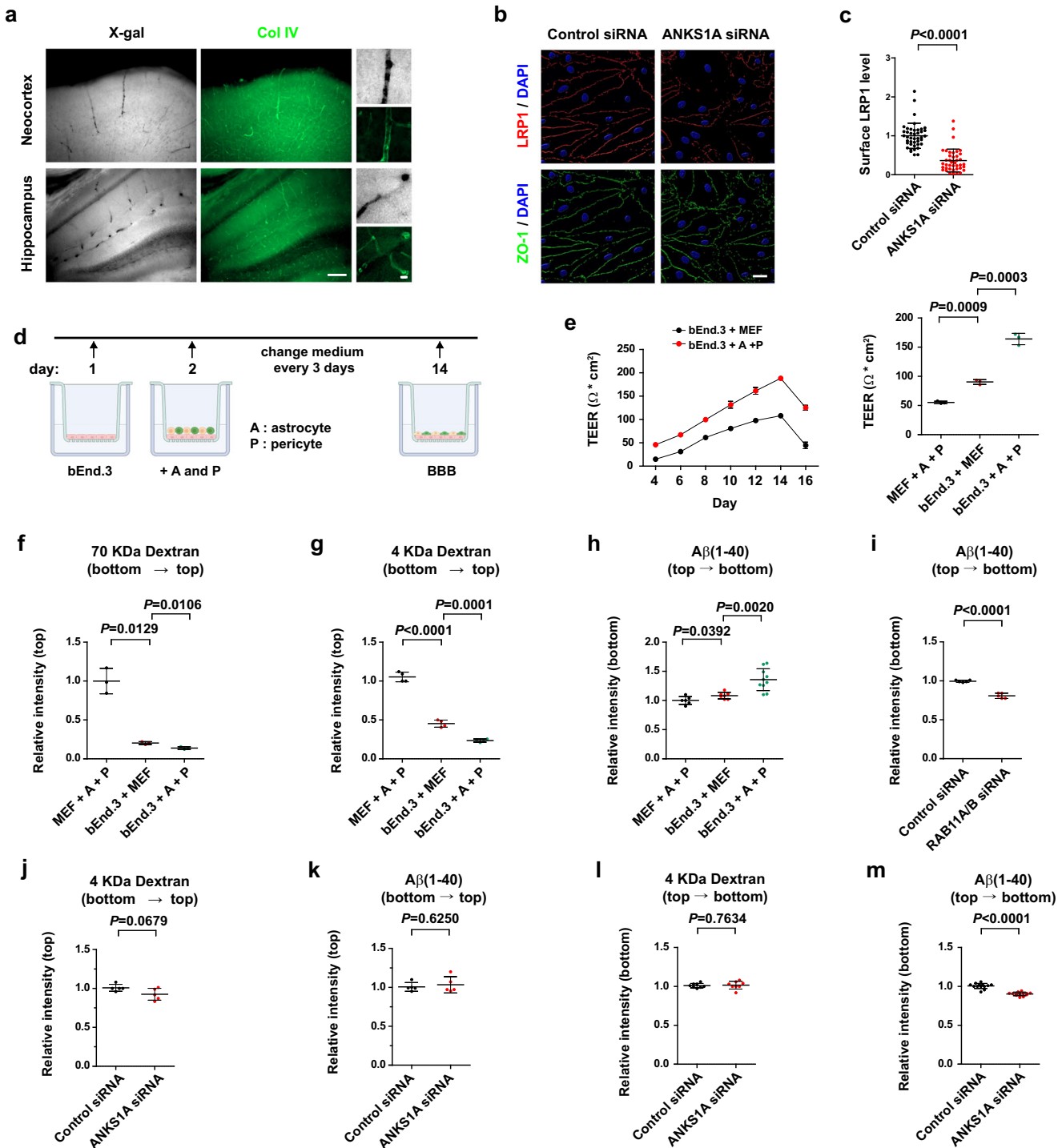

**Fig. 1 | *ANKS1A* bEnd.3 cells plays a role in the LRP1-mediated Aβ clearance.**
**a** Brain sections from *ANKS1A*[+/LacZ] mice were subjected to X-gal staining followed by a collagen (Col) IV immunohistochemical staining (scale bar, 100 μm). Results were reproduced at least five independent experiments. **b** bEnd.3 cells were transfected with the indicated siRNAs and then co-cultured with a mix of astrocytes and pericytes (scale bar, 20 μm). **c** The cell surface membrane LRP1 levels were quantified by normalizing to the intensity of ZO-1. Each dot represents the relative surface intensity of LRP1 in each microscopic field. *n* = 42 and 41 in each group from three independent experiments. **d** Schematic diagram showing the in vitro BBB preparation. The illustration in (**d**) was created with Biorender.com. **e** TEER values for the indicated BBB groups with each dot being a mean value of two readings. *N* = 3

(independent experiments) in each group. Paracellular permeability analysis of the 3 BBB groups using FITC-labeled dextran. 70 kDa dextran, *N* = 3 in each group (**f**) or 4 kDa dextran, *N* = 4 in each group (**g**). For normalization, the measured intensity of the FITC-labeled protein for the MEFs co-cultured with pericytes and astrocytes (MEF + A + P) was set to be the value of 1. **h** Transcytosis assay using Cy3-labeled Aβ(1–40). *n* = 6, 7, 10 in each group. *n* = 5 in each group (**i**–**k**). *n* = 8 in each group (**l**). *n* = 13 in each group (**m**). For normalization, the intensity of the indicated proteins in the control siRNA-transfected sample was set to the value of 1. Data shown in this figure are shown as mean ± SD. Two-tailed unpaired t-test. Source data are provided as a Source Data file.

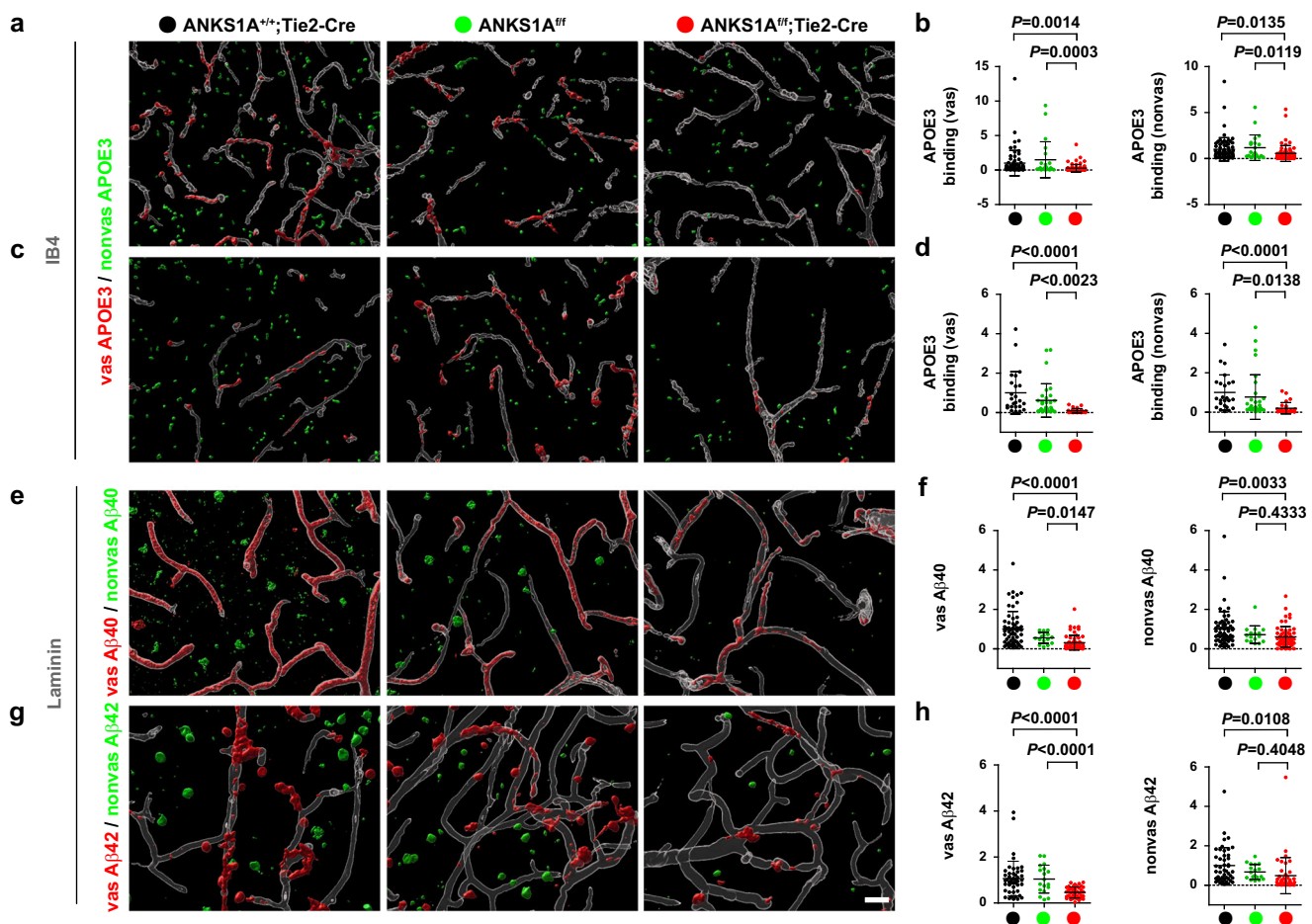

**Fig. 2 | Endothelial-specific *ANKS1A* loss results in both a decrease in cell surface LRP1 levels and a decrease in Aβ clearance.** Unfixed brains were isolated from each mouse at 2 months of age and then sliced into 200 μm-thick serial sections. The confocal images of neocortex were rendered into a 3D reconstruction using Imaris. Each section corresponding to neocortex (**a**) or hippocampus (**c**) was incubated with the indicated protein at 4 °C for overnight and then fixed prior to visualization. Vascular (vas) staining is shown as red while nonvascular (nonvas) staining is in green color. **b** Each dot represents the intensity of vascular or nonvascular staining in each microscopic field. *Tie2-Cre* group, *N* = 5 mice,

$n$ = 70 sections; *ANKS1A^{f/f}* group, *N* = 3, *n* = 22; *ANKS1A^{f/f} ;Tie2-Cre* group, *N* = 5, *n* = 74. **d** *Tie2-Cre* group, *N* = 5, *n* = 24; *ANKS1A^{f/f}* group, *N* = 3, *n* = 31; *ANKS1A^{f/f};Tie2-Cre* group, *N* = 5, *n* = 27. **e, g** Each section was incubated with Cy3-labeled Aβ(1–40) or Aβ(1–42) at 37 °C for 30 min, washed and fixed prior to visualization. Scale bar, 30 μm. **f** *Tie2-Cre* group, N = 5, *n* = 60; *ANKS1A^{f/f}* group, *N* = 3, *n* = 16; *ANKS1A^{f/f} ;Tie2-Cre* group, *N* = 5, *n* = 61. **h** *Tie2-Cre* group, *N* = 5, *n* = 46; *ANKS1A^{f/f}* group, *N* = 3, *n* = 18; *ANKS1A^{f/f};Tie2-Cre* group, *N* = 5, *n* = 41. Data in this figure are shown as mean ± SD. Two-tailed unpaired t-test. Source data are provided as a Source Data file.

endothelial LRP1 and that its loss decreases the transendothelial clearance of toxic proteins such as the amyloid peptides.

### Regulation of the brain endothelial LRP1 by ANKS1A in in vivo mouse models

We further investigated whether ANKS1A plays a role in the LRP1-mediated Aβ clearance in the mouse brain endothelium. For this purpose, we generated *ANKS1A^{f/f}; Tie2-Cre* mice[68,69]. It was observed that the conditional *ANKS1A* KO (cKO) mice appeared to be quite healthy during their lifespan, similar to the wild type (WT) mice. The cerebrovascular patterning analysis using collagen IV stained-sections did not show any significant differences between the WT and cKO group at 2 months of age (Supplementary Fig. 2a, b). Neither the pericyte-associated PDGFR-β stained sections (Supplementary Fig. 2c, d) nor the leakage analysis of peripheral blood into the brain (Supplementary Fig. 2e, f) revealed any significant differences between the WT and cKO groups. Taken together, these results suggest that *ANKS1A* loss does not perturb the development, growth, and differentiation of the cerebrovascular tissue.

To analyze whether the brain endothelial LRP1 levels would be reduced in *ANKS1A cKO* mice, freshly prepared and unfixed brain

sections were incubated with recombinant APOE3 protein at 4 °C for overnight and then washed extensively prior to visualizing the bound APOE3 (Fig. 2a). For this vascular binding analysis in the neocortex, we distinguished the vascular APOE3 binding from that of the nonvascular one: the vascular staining was defined as APOE3 puncta in direct contact with the IB4-stained vessel (see the red staining in Fig. 2a). Importantly, we found that the levels of vascular APOE3 binding were significantly reduced in *ANKS1A* cKO brains compared to control brains (Fig. 2b). This result was also observed in the hippocampus (Fig. 2c, d). In addition, the polarized expression of LRP1 to the abluminal side of endothelial cells was not misrouted to the luminal side by the ANKS1A loss (Supplementary Fig. 1g). Next, the brain tissues were treated with Aβ(1–40) peptides conjugated with a fluorescent dye at 37 °C for 30 min, which were then extensively washed prior to visualizing. As expected, the levels of vascular Aβ(1–40) peptide binding were significantly reduced in *ANKS1A* cKO brains compared to control brains (Fig. 2e, f and Supplementary Fig. 2h, i). We also observed a similar result for the Aβ(1–42) peptide binding in the neocortex, although the LRP1 internalization effect was mild or not significant for Aβ(1–42) peptides in the hippocampus (Fig. 2g, h and Supplementary Fig. 2j, k). Taken together, our results strongly suggest

that ANKS1A plays a role in regulating the homeostatic levels of the brain endothelial LRP1. Interestingly, we observed that nonvascular binding of APOE3 but also nonvascular internalization of Aβ peptides were slightly reduced in *ANKS1A* cKO brains, although this reduction varied depending on the brain regions studied (Fig. 2b, d, f, h and Supplementary Fig. 2i, k). However, the level of mature LRP1 in non-vascular tissues was not reduced in *ANKS1A* cKO mice compared with control mice such as *Tie2-Cre* or *ANKS1A^f/f^*(Supplementary Fig. 2l, m). The level of ANKS1A in non-vascular tissues was barely detectable in control or *ANKS1A* cKO mice. In addition, *Tie2-Cre*; *Ai9* mice showed specific expression of tdTomato in brain endothelial cells but not in other parenchymal cells[70] (Supplementary Fig. 2n). More importantly, the level of mouse APOE was significantly increased in the nonvascular tissues of *ANKS1A* cKO mice compared with control mice (Supplementary Fig. 2l, m). These results suggest that due to ineffective cerebrovascular clearance, the nonvascular parenchymal tissues had accumulated levels of LRP1 ligands, affecting the binding of APOE3 to the nonvascular LRP1 protein.

We further investigated whether *ANKS1A* loss leads to the vascular build-up of Aβ peptides in the brain in an AD mouse model. For this purpose, we crossed the *ANKS1A* cKO mice with *5XFAD* mice, which express human *APP* and *PSEN1* transgenes, having a total of five AD-linked mutations[71,72]. Sex differences have been reported in *5XFAD* mice, including Aβ pathology being higher in female than in male mice[73–76]. To control for this gender difference in the Aβ pathological analysis, we focused on male mice for the remainder of the experiments. In the crossed mice samples, we scored any Aβ-stained puncta as being cerebral amyloid angiopathy (CAA) when they were within 5 μm from the collagen IV-stained vessel (see Supplementary Movie 1). The number of CAAs in *5XFAD* mice at 5 months of age was increased, correlating with *ANKS1A* loss and that this trend of Aβ build-up was observed both in the neocortex and the hippocampus (Supplementary Fig. 3a, b). In addition, the collagen IV-based vascular patterning analysis indicated that *ANKS1A* loss did not have a significant effect on the vascular vessel density, average length of the vessels, or the gaps between the vessels in both the neocortex and the hippocampus, although its loss had a marginal effect on the branching index in the mice samples (Supplementary Fig. 3c–f).

To further test whether the vascular Aβ build-up in *ANKS1A* cKO;*5XFAD* mice disrupts the integrity of the cerebrovascular system, a similar analysis was performed on samples from the 7-month-old mice. Indeed, for these animals, the total plaque coverage analysis revealed that *ANKS1A* deficiency resulted in an increase in overall Aβ plaque pathology in their *5XFAD* background (Supplementary Fig. 3g, h). Importantly, the endothelial-specific *ANKS1A* loss in the *5XFAD* background resulted in exacerbating not only the vascular Aβ build-up (Fig. 3a, b), but also disturbed cerebrovascular patterning (Fig. 3c, d, and Supplementary Fig. 3i). In addition, *ANKS1A* cKO brains had more nonvascular parenchymal Aβ accumulated, most likely due to their decreased cerebrovascular Aβ clearance (Fig. 3b, fourth panels). These findings strongly suggest that *ANKS1A* loss induces a progressive cerebrovascular breakdown through impaired Aβ clearance and consequent toxic amyloid accumulation. The 7-month-old *5XFAD* mice already showed a significant leakage of peripheral blood into the brain tissue and this BBB leakage was similar to the levels we observed in *ANKS1A* cKO;*5XFAD* mice (Supplementary Fig. 3j). This suggests that 7-month-old *5XFAD* mice are rather complicated for studying the effects of *Tie2-Cre* mediated *ANKS1A* ablation on BBB integrity.

## Mechanism underlying the regulation of endothelial LRP1 by ANKS1A

To elucidate the molecular mechanism underlying ANKS1A regulation of the LRP1-dependent Aβ clearance, we generated various C-terminal mutants of the human LRP1 minireceptor (mLRP4-T100), also expressing an N-terminal HA tag[44] (Fig. 4a). HEK293 cells express a moderate level of ANKS1A (Supplementary Fig. 4b, lane 1), but cell surface membrane staining with anti-HA antibody allowed us to examine only their ectopically expressed recombinant LRP1 minireceptor (Fig. 4b, d). In two independent *ANKS1A* KO cell lines generated by CRISPR/Cas9 editing (Supplementary Fig. 4a, b), the cell surface levels of ectopic LRP1 were markedly reduced (Fig. 4b, c, lanes 2, 4), whereas those of the ectopic LRP1 were restored to control levels by the ANKS1A expression (lanes 3, 5). We also confirmed binding of the LRP1 cytoplasmic tail GST fusion protein to ANKS1A His-tagged PTB fusion protein, demonstrating a direct binding of ANKS1A to the LRP1 C-terminal domain (Supplementary Fig. 4c). Using various C-terminal mutants of LRP1 (Fig. 4a), we found that two NPXY motifs in the LRP1 cytoplasmic tail required ANKS1A for its proper localization to cell surface (Fig. 4d, e). In particular, the mutant, N26A/N60A, in which the asparagine was replaced with alanine at each NPXY motif, displayed the least localization of LRP1 to cell surface membrane (lane 5). The L66A mutant of the YXXL motif not only remained at the surface, it was more abundantly so than the wild-type minireceptor (lane 6), consistent with the previous reports that the YXXL motif is critical for the internalization of LRP1[18,44].

Our previous study showed that ANKS1A is specifically localized to the ER and that it regulates COPII-mediated anterograde transport of specific cargoes[62], raising the possibility that LRP1 is dependent on ANKS1A for its ER export. We therefore examined whether the LRP1 minireceptor is properly processed in the ER of HEK293 cells. Western blot analysis showed that two distinct bands, 120 kDa and 200 kDa in size, detected by anti-HA antibody probing of the cell lysates (Fig. 4f). The 200 kDa band represents the full-length ER precursor form of the LRP1 minireceptor, whereas the 120 kDa represents the processed form of the receptor in the Golgi[44]. Importantly, the relative levels of the minireceptor retained in the ER were much higher for the NPXY mutants than for the wild type or the YXXL mutant (Fig. 4g). Immunostaining analysis of the cells using ER-specific markers confirmed that the LRP1 minireceptor NPXY motif mutant was more abundantly detected in the ER (Fig. 4h, i and Supplementary Fig. 4d, e). In addition, bEnd.3 cells, at least 16% of ANKS1A was found in the ER compartment while 7% of ANKS1A in the Golgi (Supplementary Fig. 4f, g). In bEnd.3 cells with knockdown of *ANKS1A*, LRP1 was more highly co-localized with the ER-specific markers (Fig. 4j, k). More importantly, our co-immunoprecipitation experiments confirmed the specific interaction of ANKS1A with endogenous LRP1 in primary brain endothelial cells co-cultured under a mixture of astrocytes and pericytes in a transwell format (Fig. 4l–n). Taken together, these results strongly indicate that ANKS1A interacts with the NPXY motifs of the properly folded LRP1 in the ER and that it promotes the anterograde transport of LRP1 from the ER.

## Rescue of the defective vascular clearance by the brain endothelial *ANKS1A* gene delivery

In our study, endothelial-specific *ANKS1A* loss did not exert its adverse effect on the cerebrovascular patterning of *5XFAD* mice before 7 months of age, so we used the conventional *ANKS1A* KO mice[62,77] with the genetic background of *5XFAD* to test whether these mice displayed the severe cerebrovascular pathology at 5 months of age. Around the age of 5 months, *5XFAD* mice do not display signs of BBB leakage. It was noted that with *ANKS1A* KO in the *5XFAD* background, the mice exhibited a more accelerated vascular Aβ pathology than the endothelial-specific *ANKS1A* cKO;*5XFAD* mice. Both Aβ puncta and CAA sizes were markedly increased (Fig. 5a, b); in addition, the vascular patterning was disturbed along with an increased leakage of peripheral blood into parenchymal tissues (Supplementary Fig. 5a, b, g, h). From *ANKS1A* loss in *5XFAD* male mice, this severe Aβ pathological symptom was also found in nonvascular parenchymal tissue and was likely due to decreased cerebrovascular Aβ clearance (Fig. 5b, third panels). Similar Aβ phenotypes were also observed in *5XFAD* female mice

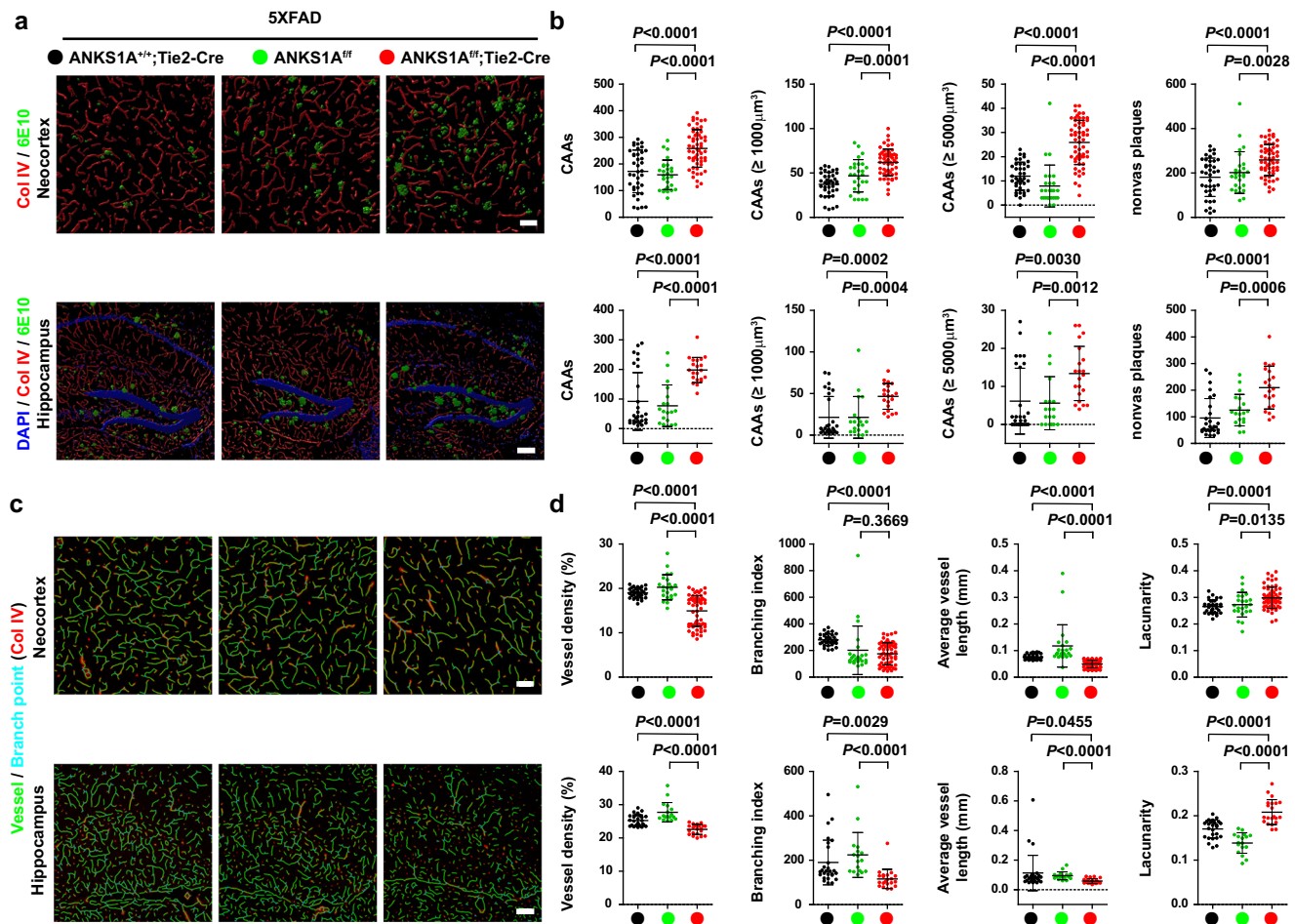

**Fig. 3 | Endothelial-specific *ANKS1A* loss exacerbates both the vascular Aβ build-up and the abnormal vascular patterning in *5XFAD* mice.** Fixed brains were prepared from each mouse at 7 months of age. **a** Immunohistochemical staining with 6E10, the monoclonal antibody against the Aβ peptides, and the Col IV-specific antibody. Vascular 6E10-positive plaques (CAAs) (within 5 μm from the vessel) are shown in green (scale bars, 50 μm (neocortex), 100 μm (hippocampus)). **b** For quantification, the number of vascular plaques (CAAs) or nonvascular plaques were measured in each field. Each dot represents the number of amyloid deposits in each microscopic field (*Tie2-Cre* group, *N* = 5 mice, *n* = 38 sections; *ANKS1A^{f/f}* group, *N* = 3, *n* = 27; *ANKS1A^{f/f}*;*Tie2-Cre* group, *N* = 5, *n* = 57 for upper panels; *Tie2-Cre* group, *N* = 5, *n* = 28; *ANKS1A^{f/f}* group, *N* = 3, *n* = 19; *ANKS1A^{f/f}*;*Tie2-Cre* group, *N* = 5, *n* = 21 for lower panels). **c** Immunohistochemical staining using Col IV-specific antibodies.

The images were rendered into the vascular network images using the AngioTool software: the vessel skeletons (green) and vascular branching points (sky blue) were merged with the Col IV-stained image (red) (scale bars, 50 μm (neocortex), 100 μm (hippocampus)). **d** Values for vessel density, branching points, average vessel length and lacunarity (meaning gap in spatial pattern) were measured using AngioTool. Each dot represents the value in each microscopic field (*Tie2-Cre* group, *N* = 5, *n* = 32; *ANKS1A^{f/f}* group, *N* = 3, *n* = 23; *ANKS1A^{f/f}*;*Tie2-Cre* group, *N* = 5, *n* = 61 for upper panels; *Tie2-Cre* group, *N* = 5, *n* = 27; *ANKS1A^{f/f}* group, *N* = 3, *n* = 17; *ANKS1A^{f/f}*;*Tie2-Cre* group, *N* = 5, *n* = 21 for lower panels). Data in this figure are shown as mean ± SD. Two tailed unpaired t-test against *ANKS1A^{f/f}* ; *Tie2-Cre* populations. Source data are provided as a Source Data file.

(Supplementary Fig. 5c–f). Interestingly, with heterozygous *ANKS1A* loss in the *5XFAD* female background, we also observed that the mice manifested a more accelerated vascular Aβ pathology than control *5XFAD* female mice. We further assayed the levels of Aβ(1–40) and Aβ(1–42) in brain microvessel lysates via ELISA to investigate the characteristics of the deposited amyloid peptides. In agreement with the previous immunohistochemistry results, *ANKS1A* deficiency significantly increased both Aβ(1–40) and Aβ(1–42) levels in Tris-soluble or FA-soluble cerebral microvessel homogenates (Fig. 5c, lanes 1 and 2), demonstrating that ANKS1A is critical for regulating the cerebrovascular clearance of Aβ peptides.

We next used a brain endothelial-specific viral vector, *AAV2-BRI*[78], for delivery of *ANKS1A* and to treat the severe vascular Aβ pathology in *ANKS1A* KO;*5XFAD* mice. Consistent with the previous report, *AAV2-BRI-GFP* viral injection via tail vein revealed remarkable specificity and long-lasting expression of GFP in brain endothelial cells until at least six months post viral injection (Supplementary Fig. 5i). With the *AAV2-BRI-ANKS1A* virus therapeutic candidate, a single intravenous injection

at 2 months of age was performed. The injected *ANKS1A* KO;*5XFAD* mice were analyzed at 5 months of age. Interestingly, we found no significant reduction of Aβ(1–40) and Aβ(1–42) levels in Tris-soluble fractions in the injected group (Fig. 5c, first and second panels, lane 3). However, for the FA-soluble fractions, representing the insoluble fraction, the therapeutic viral vector expressing *ANKS1A* markedly attenuated the Aβ(1–40) and Aβ(1–42) levels; in particular, the levels of Aβ(1–42), a highly neurotoxic, predominant component in the *5XFAD* brain (third and fourth panels, lane 3), were reduced. These suggested that the viral expression of *ANKS1A* was more effective in preventing Aβ peptides from converting into an insoluble structure in the cerebrovascular system.

To further investigate whether the various Aβ pathology patterns in the three *5XFAD* groups correlated with the degree of cognitive behavioral deficit, we performed a modified version of the Y-maze task on these mice; this test measures spatial reference memory[79]. In the Y-maze test, the *ANKS1A* KO;*5XFAD* mice showed reduced entry frequency and time spent in the novel arm compared to control *5XFAD*

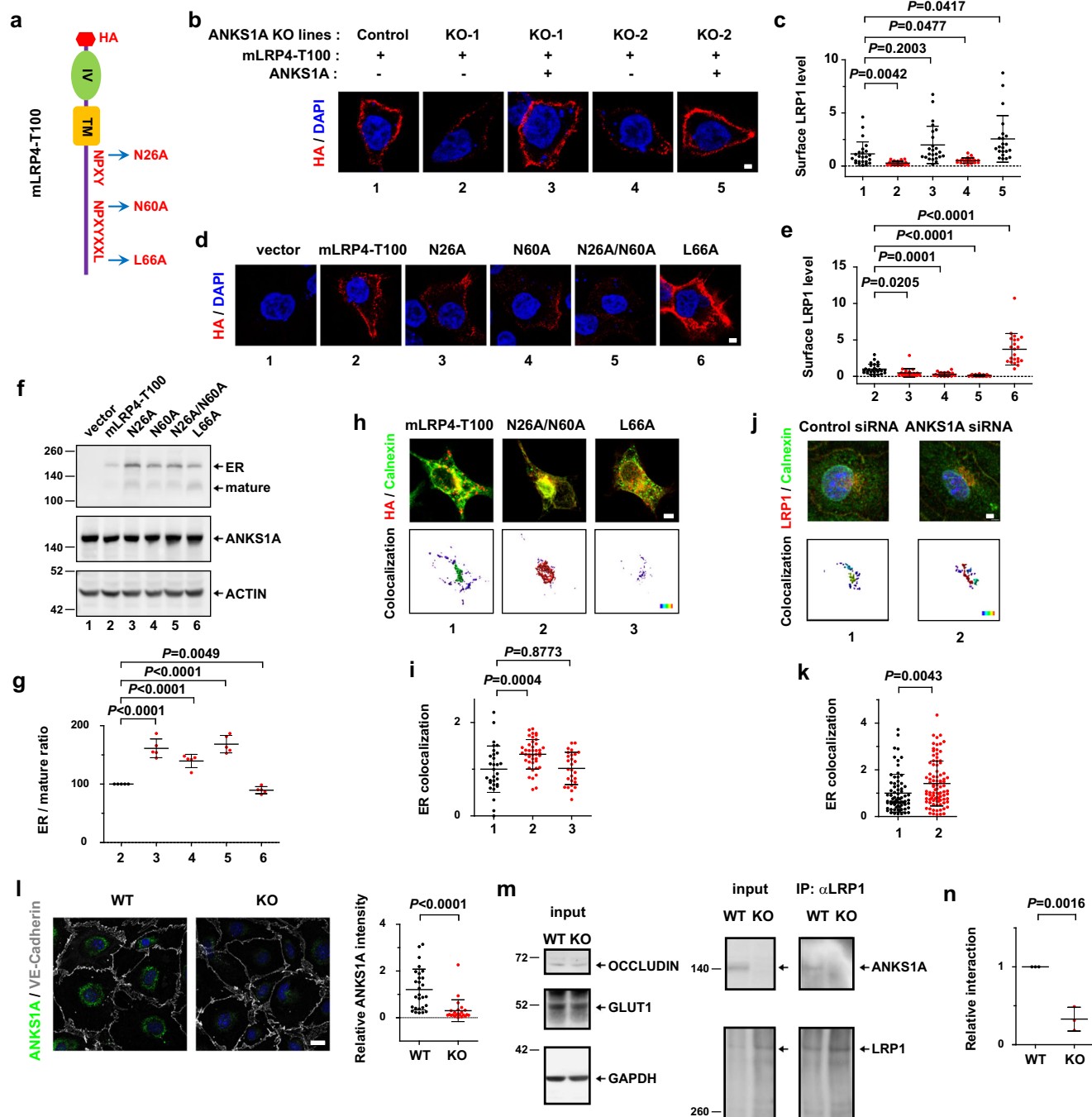

**Fig. 4 | ANKS1A interacts with the NPXY motifs of LRP1 to promote its export from the ER. a** Schematics showing the human LRP1 minireceptor (mLRP4-T100). HA hemagglutinin; TM transmembrane domain; IV ligand binding region 4. **b, d** Transfected HEK293 cells were stained with anti-HA antibody and confocal images are shown (scale bar, 2 μm). **c** Each dot represents the intensity of the surface-localized minireceptor per cell. $n$ = 24 cells (group 1–4) and 22 cells (group 5). **e** $n$ = 27, 24, 23, 27, and 22 cells for each group. **f** Western blot analysis using the lysates from cells in Fig. 4c. **g** Each dot represents a ratio of the 200 kDa ER LRP1 form to the 120 kDa cleaved form. $N$ = 5 independent experiments for each group. **h** The minireceptor expression in HEK293 cells was analyzed with anti-HA and anti-calnexin antibodies (scale bar, 5 μm). **i** "% of ROI colocalized" in the colocalization channel was used to obtain the ER colocalization coefficient. The mean value of the WT minireceptor-expressing cells was set to the value of 1 for normalization. $n$ = 31,

42, and 27 cells for each group. **j, k** siRNA-transfected bEnd.3 cells were stained with LRP1-specific and ER-marker antibodies (scale bar, 2 μm). $n$ = 71, 90 cells for each group. **l** In a transwell setup, primary brain endothelial cells were co-cultured with primary pericytes and astrocytes (scale bar, 20 μm). $n$ = 28, 24 cells for each group. **m** Western blot analysis of primary brain endothelial cell extracts. The cell lysates were incubated with the anti-LRP1 antibody for co-immunoprecipitation. The ER precursor form of LRP1 (~530 kDa in size) was resolved in a gradient gel. **n** For quantification, ANKS1A levels were normalized with those of LRP1. $N$ = 3 for each group. The rainbow scale (**h, j**) represents the low (violet) to high (red) co-localization (colormap means statistics coded for intensity sum of fluorescences, $1.0 \times 10^4$ - $1.0 \times 10^8$ spectrum). Data (**c, e, i, k, l, n**) were from three independent experiments. Data in this figure are shown as mean ± SD. Two tailed unpaired t-test against control or WT populations. Source data are provided as a Source Data file.

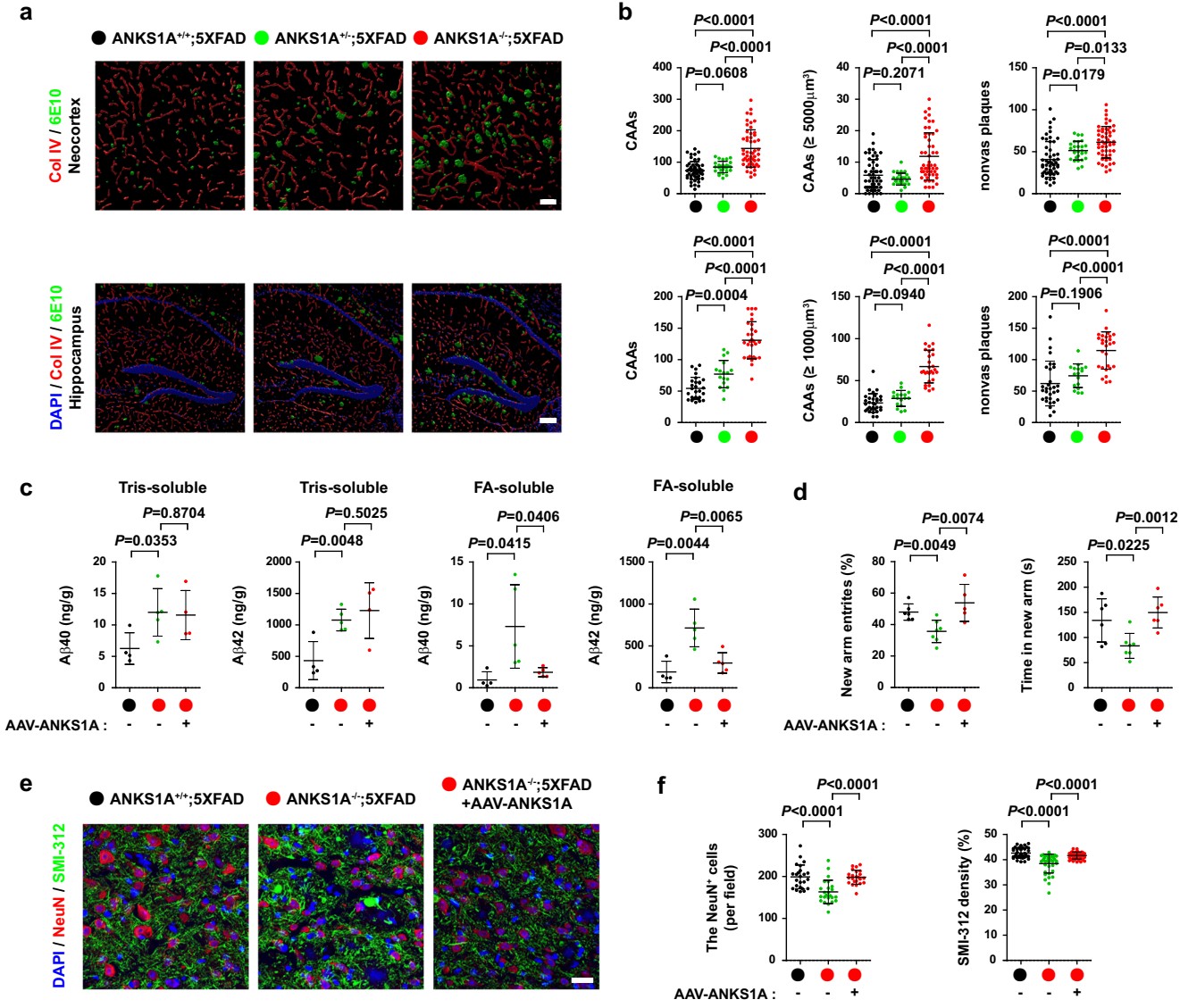

**Fig. 5 | *ANKS1A* deficiency exacerbates the AD-related pathology in *5XFAD* mice and its adverse effect is reversed by *ANKS1A* gene delivery.** Mice used in the experiment were 5 months old. **a, b** *5XFAD* group, *N* = 5 mice, *n* = 53 sections; *ANKS1A*[+/−]*;5XFAD*, *N* = 3, *n* = 27; *ANKS1A*[−/−]*;5XFAD*, *N* = 5, *n* = 52 for upper panels; *5XFAD*, *N* = 5, *n* = 28; *ANKS1A*[+/−]*; 5XFAD*, *N* = 3, *n* = 17; *ANKS1A*[−/−]*;5XFAD*, *N* = 5, *n* = 28 for lower panels (scale bars, 50 μm (neocortex), 100 μm (hippocampus)). **c** *AAV-BR1-ANKS1A* virus was injected into the tail vein of each mouse at 2 months of age, and the injected mice were sacrificed at 5 months of age. Microvessel homogenates were separated into TBS (Tris-based) soluble and FA (formic acid) soluble fractions. Each fraction was used for measuring the levels of Aβ40 or Aβ42 via ELISA. For normalization, the levels of the Aβ peptides measured by ELISA were divided by the weight of isolated microvessels for each animal. *ANKS1A*[+/+]*;5XFAD*, *N* = 4; *ANKS1A*[−/−]*;5XFAD*, *N* = 5; *ANKS1A*[−/−]*;5XFAD + AAV-ANKS1A*, *N* = 4 (Tris-soluble) or 5 (FA-soluble). **d** The modified Y-maze test used for cognitive behavior of each animal. Each dot represents the behavioral score for each animal. *ANKS1A*[+/+]*;5XFAD*, *N* = 6; *ANKS1A*[−/−]*;5XFAD*, *N* = 7; *ANKS1A*[−/−]*;5XFAD + AAV-ANKS1A*, *N* = 6. **e, f** The NeuN-positive neurons in the neocortical layer V were analyzed (scale bar, 20 μm). Each dot represents the number of NeuN-positive neurons or the SMI312-positive neuritic density per field (*N* = 3 per group; *n* = 24, 24, 23 for each group (NeuN); *n* = 37, 37, 40 for each group (SMI312)). Data in this figure are shown as mean ± SD. Two-tailed unpaired t-test against *ANKS1A*[−/−]*;5XFAD* populations. Source data are provided as a Source Data file.

mice (Fig. 5d, lanes 1 and 2); however, this severe memory impairment was significantly rescued by the *AAV2-BR1-ANKS1A* viral injection (lane 3). In addition, the *AAV2-BR1-ANKS1A* virus injected group consistently had increased numbers of NeuN[+] neurons and SMI312[+] neurites both in the cortex and hippocampus as compared with *AAV2-BR1-GFP* injected group (Fig. 5e, f and Supplementary Fig. 5l, m). In contrast, WT mice injected with *the AAV2-BR1-ANKS1A* virus displayed normal cognitive behavior and had normal NeuN[+] neuron numbers (Supplementary Fig. 5j, k). Taken together, our results suggest that the vascular clearance impairment of Aβ peptides results in accumulation of both soluble and insoluble Aβ peptides and that the buildup of insoluble Aβ peptides is more related to neurodegenerative and cognitive impairments. Furthermore, the brain endothelial-specific *ANKS1A* gene delivery has implications for treatment of neurodegeneration linked to vascular dysfunction.

## Cerebral vascular clearance in human iPSC endothelium

A recent study showed that an in vitro iPSC-based three-dimensional model of human BBB recapitulates the physiological properties of the BBB[80]. We next used this BBB in vitro model to address whether *ANKS1A* loss results in the vascular clearance dysfunction in human BBB models. We first used a well-established protocol[80] for each cell type for differentiating iPSCs into brain endothelial cells (iECs), pericytes (iPCs), and astrocytes (iASs), and then validated the identity of each differentiated cell using RNA sequencing, RT-qPCR, and specific marker expression along with confirming the cell's known

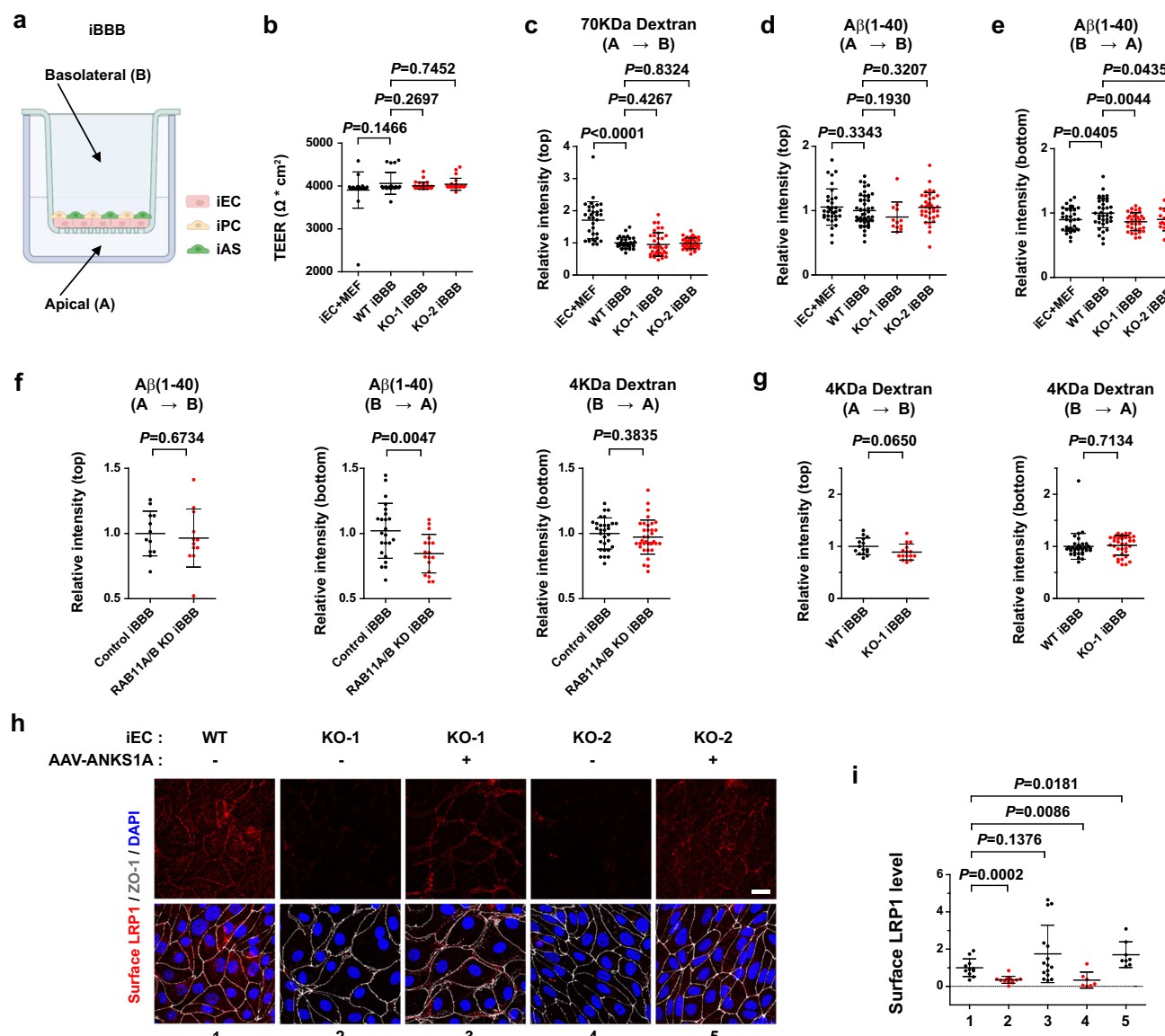

**Fig. 6 | *ANKS1A* loss in human iPSCs impairs transendothelial clearance through decreasing the cell surface levels of LRP1.** The human iPSC-derived cells used in this figure were generated from the GM23720 cell line. **a** Schematic diagram showing the iBBB setup, where iPSC-derived endothelial cells (iECs) were cultured for one day prior to being overlaid by the mix of iPSC-derived pericytes (iPC) and astrocytes (iASs). The iBBBs were cultured for 14 days. The illustration in panel a was created with Biorender.com. **b** TEER measurements for each cellular barrier model. *n* = 22, 22, 30, and 23 independent wells for each group. **c, d** Paracellular permeability assay using FITC-labeled 70 kDa dextran (*n* = 33, 36, 36, 36 for each group) or Cy3-labeled Aβ(1–40) (*n* = 34, 48, 12, 36 for each group). **e, f** The transcytosis assay using Cy3-labeled Aβ(1–40) peptide (**e**, *n* = 33, 36, 33, 35 for each group; **f**, first panel, *n* = 12 for each group, second panel, *n* = 23, 18 for each group, third panel, *n* = 30, 35 for each group. **g** *n* = 15 for each group (first panel); *n* = 36 for each group (second panel). **h** iECs grown on the dish were incubated with *AAV-BR1-ANKS1A* virus (incubated with 10^10 genome copy number (gcn)/ml) for overnight and then co-cultured with the transwell insert containing the mix of iPCs and iASs. The iECs were then stained with the LRP1- and ZO-1 specific antibodies for cell surface LRP1 (scale bar, 20 μm). **i** Data in Fig. 6h were quantified with WT iECs values set to 1 for relative surface LRP1 levels in each cell group. *n* = 11, 13, 15, 7 for each group from three independent experiments. Data in this figure are shown as mean ± SD. Two tailed unpaired t-test against WT iBBB populations. Source data are provided as a Source Data file.

morphology[81–84] (Supplementary Fig. 6a, b and Supplementary Table 1). We next established a transwell system for a confluent culture of iECs in matrigel followed by being overlaid by pericytes and astrocytes (Fig. 6a). The iPSC-derived ECs cultured with MEFs showed a high TEER with an average of 4000 ohm•cm² (Fig. 6b, lane 1) with the 14-day co-culture. TEERs for the iECs co-cultured with pericytes and astrocytes, termed as iBBB, exhibited a maximal TEER plateau (4200 ohm•cm²) after the 14-day co-culture (Fig. 6b, lane 2 and Supplementary Fig. 6c). We next examined the paracellular permeability of the 70 kDa dextran, revealing that the iBBB had a decreased permeability of 50% compared to the iECs co-cultured with MEFs (Fig. 6c, lanes 1, 2).

We also generated isogenic *ANKS1A* KO iPSC cell lines using the CRISPR/Cas9 editing system and then used these KO cells for differentiating them into iECs (Supplementary Fig. 6d–f). In iECs, the levels of receptor-associated protein[85–87], a molecular chaperone for LRP1, were not changed (Supplementary Fig. 6g). In the iBBB derived from the *ANKS1A* KO iECs, we did not find any significant changes in the average TEER or paracellular permeability compared to iBBB from *ANKS1A* WT iECs (Fig. 6b, c, lanes 3, 4). To further examine whether the established iBBB can model the LRP1-mediated vascular clearance of the Aβ peptides, the iBBB was first exposed to Aβ(1–40) peptide in the apical side for 1 h, showing that both *ANKS1A* WT and KO iBBB had a

similar paracellular permeability as compared with the iECs co-cultured with MEFs (Fig. 6d). However, in similar experiments where Aβ(1–40) peptide was added to the abluminal side, WT iBBB exhibited 12% more increased permeability compared to the iECs co-cultured with MEFs (Fig. 6e, lanes 1, 2). This increased permeability of Aβ(1–40) peptide was effectively inhibited by the knock-down of *RAB11A* and *RAB11B*, the small GTPase paralogs critical for transcytosis, while the paracellular permeability of the 4 kDa dextran was not affected (Fig. 6f and Supplementary Fig. 6h). This strongly suggested that the iBBB displays the abluminal polarization of endothelial LRP1 and that it can transport the Aβ peptides from its abluminal to its apical side. Importantly, for *ANKS1A* KO iBBB, the increased transport of Aβ(1–40) peptide from the abluminal to the apical side was abrogated (Fig. 6e, lanes 3, 4). A similar result was also observed in iBBBs with the Aβ(1–42) peptide treatment (Supplementary Fig. 6i). As a control, the permeability of the 4 kDa dextran in iBBBs was not altered by *ANKS1A* KO iECs (Fig. 6g). In similar experiments using the iBBB containing *ANKS1A* KO pericytes or astrocytes, we did not observe any changes to iEC-dependent transcytosis of Aβ(1–40) peptide (Supplementary Fig. 6j). Consistent with these iBBB results, there was specific binding of Aβ(1–40) peptide to cell surface of iECs at 4 °C, followed by internalization at 37 °C (Supplementary Fig. 6k). Furthermore, Western blot analysis of the *ANKS1A* KO iEC lysates revealed the endogenous LRP1 precursor form being more abundantly present in the ER of the cells (Supplementary Fig. 6l). We also confirmed that *AAV2-BR1-GFP* viral transduction occurs more dominantly in iECs than other cell types (Supplementary Fig. 6m) and that the anti-LRP1 antibody specifically recognizes cell surface-bound LRP1 in iECs (Supplementary Fig. 6n, o). Notably, the cell surface levels of LRP1 in iECs were significantly decreased with *ANKS1A* KO (Fig. 6h, i, lanes 2, 4,) but were restored to normal levels by *AAV2-BR1-ANKS1A* viral transduction (lanes 3, 5). Taken together, from our models of study, the results strongly suggest that *ANKS1A* loss is a risk factor for cerebral amyloid angiopathy (CAA) in humans.

### Impaired cerebrovascular clearance of the *rs6930932 ANKS1A* missense variant

No known AD-associated single-nucleotide polymorphisms (SNPs) in *ANKS1A* have been identified up to date. To discover any potential AD-associated *ANKS1A* mutations, we analyzed the exome sequencing data of the Korean populations with early onset Alzheimer's disease (EOAD) that develops before the age of 65 years[73] (*n* = 39, Supplementary Table 2). Among 51 SNPs recorded, the *rs820085*, *rs71538280*, and *rs6930932* variants were predicted to have moderate impact mutations as missense variants (*rs820085* and *rs6930932*) or an insertion (*rs71538280*). Next, we compared the frequency of these SNPs to those found in the reference population of the 1000 Genome Project (GP)[74,75], and the Genome Aggregation Database (gnomAD) v.2.1.1[88]. We found that the frequency of *rs6930932* in our 39 samples was 5–10 times higher than in East Asian populations in the 1000 GP and gnomeAD, and the Korean in gnomeAD sets (frequencies 0.051 vs. 0.011, 0.051 vs. 0.005, and 0.051 vs. 0.008, respectively; Supplementary Data 1). Each patient with the *rs6930932* SNP carries a heterozygous p.A355D missense mutation in ANKS1A. The Ala 355 in human ANKS1A is highly conserved among mammals and is located in a structurally uncharacterized region between Ankyrin repeats and its first SAM domain (Supplementary Fig. 7a). To test any functional differences between the WT and the A355D variant, we expressed mLRP4-T100 in HEK293 cells and compared its cell surface localization changes in presence of WT ANKS1A or its A355D variant (Fig. 7a, b). Unlike control cells, the *ANKS1A* KO cells did not display an ectopic LRP1 staining on the cell surface membrane; this effect, however, was rescued by WT ANKS1A expression, but not with the A355D mutant (Fig. 7a, b, lanes 2–4 and Supplementary Fig. 7b). In addition, we found that the ectopic LRP1 was more prominently retained in the ER of *ANKS1A* KO and

*A355D*-transfected KO cells (Fig. 7a, b, lanes 2, 4). These results demonstrate that the A355D variant of ANKS1A is a loss-of-function mutant.

We next used human isogenic *ANKS1A* KO iPSC lines to confirm the loss-of-function properties of the A355D variant. The *ANKS1A* KO iECs had a low level of LRP1 on the cell surface membrane as compared to WT iECs (Fig. 7c, d, lanes 1, 2); this KO phenotype was clearly rescued by WT *ANKS1A* viral transduction (lane 3). More importantly, this phenotype was not rescued by *ANKS1A A355D* viral transduction (lane 4), and LRP1 was more detectable in the ER in *ANKS1A* KO iECs and *A355D*-transduced KO iECs (lanes 2, 4).

To further examine whether the ANKS1A A355D variant affects the cerebrovascular clearance, we generated iBBBs from isogenic iPSC lines carrying the *rs6930932* SNP variant (Fig. 7e, f and Supplementary Fig. 7c, d). Importantly, *A355D* heterozygous (Het) iECs and homozygous (Hom) iECs revealed a low level of LRP1 on the cell surface membrane as compared to WT iECs (Fig. 7g, h). In addition, immunostaining analysis revealed that the extent of A355D localized with the ER was increased in an allele dose-dependent manner (Fig. 7i, j). Next, we performed a comparative analysis of three iBBBs derived from isogenic WT, Het and Hom iECs for the *ANKS1A* A355D mutation: both iPCs and iASs used for generating iBBBs had a normal *ANKS1A* gene. The iBBBs containing *ANKS1A A355D* iECs did not show significant changes in TEER or paracellular permeability of the 70 kDa dextran or Aβ(1–40) peptides as compared with WT iBBBs (Supplementary Fig. 7e). However, *ANKS1A A355D* Het and Hom iBBBs exhibited significantly lower levels of the Aβ(1–40) peptide in the lower chamber of the transwell assay (Fig. 7k). We also obtained a similar result using iBBBs derived from a different iPSC cell line and its isogenic lines carrying the *ANKS1A* mutations, suggesting that our data is unlikely to be the result of clonal variation in the genetic editing process (Supplementary Fig. 7f–h). Taken together, these results strongly suggest that the *ANKS1A A355D* variant is potentially a risk factor for increased AD pathogenesis and other vascular pathologies.

## Discussion

In this study, we found that *ANKS1A* deficiency in the brain endothelium correlates with reduced cell surface levels of LRP1, followed by inefficient Aβ clearance in the various BBB models. At a molecular level, we found that ANKS1A binding to the NPXY motifs in LRP1 enhances the export of LRP1 from the ER. In the AD mouse model, *ANKS1A* deficiency exacerbated Aβ deposition in the brain endothelium, BBB breakdown and cognitive impairment due to the impaired cerebrovascular Aβ clearance. Deficiency and re-expression of *ANKS1A* in endothelial cells clearly rescued Aβ clearance in the various BBB models and prevented insoluble Aβ build-up in the mouse brain and consequent cognitive behavioral impairments. Using human iPSC-derived endothelial cells and iBBBs, we found that the *rs6930932* variant encoding ANKS1A A355D is a loss-of-function mutation for the anterograde transport of LRP1 from the ER and that its endothelial expression results in decreasing Aβ clearance across the iBBBs. Collectively, our findings indicate that ANKS1A regulates LRP1-mediated Aβ clearance across the BBB and a reduced ANKS1A function is a risk factor for CAA-associated AD pathogenesis (Supplementary Fig. 8).

A leading hypothesis for the cause of AD is the "Amyloid Cascade Hypothesis," suggesting that imbalances in Aβ amyloid peptide metabolism enhance the accumulation and neurotoxic aggregation of Aβ peptides in the brain, which in turn initiate neurodegeneration and cognitive impairments[2]. Abnormal accumulation of Aβ peptides from an impaired elimination of Aβ from the brain results in the hallmark formation of insoluble fibrils and senile plaques for AD, resulting in neuronal damage[1]. Of the multiple clearance systems to remove the extracellular Aβ peptides from the brain, the LRP1-mediated cerebrovascular Aβ clearance is foremost among them as an inefficient LRP1-mediated endothelial clearance has been shown to lead severe Aβ

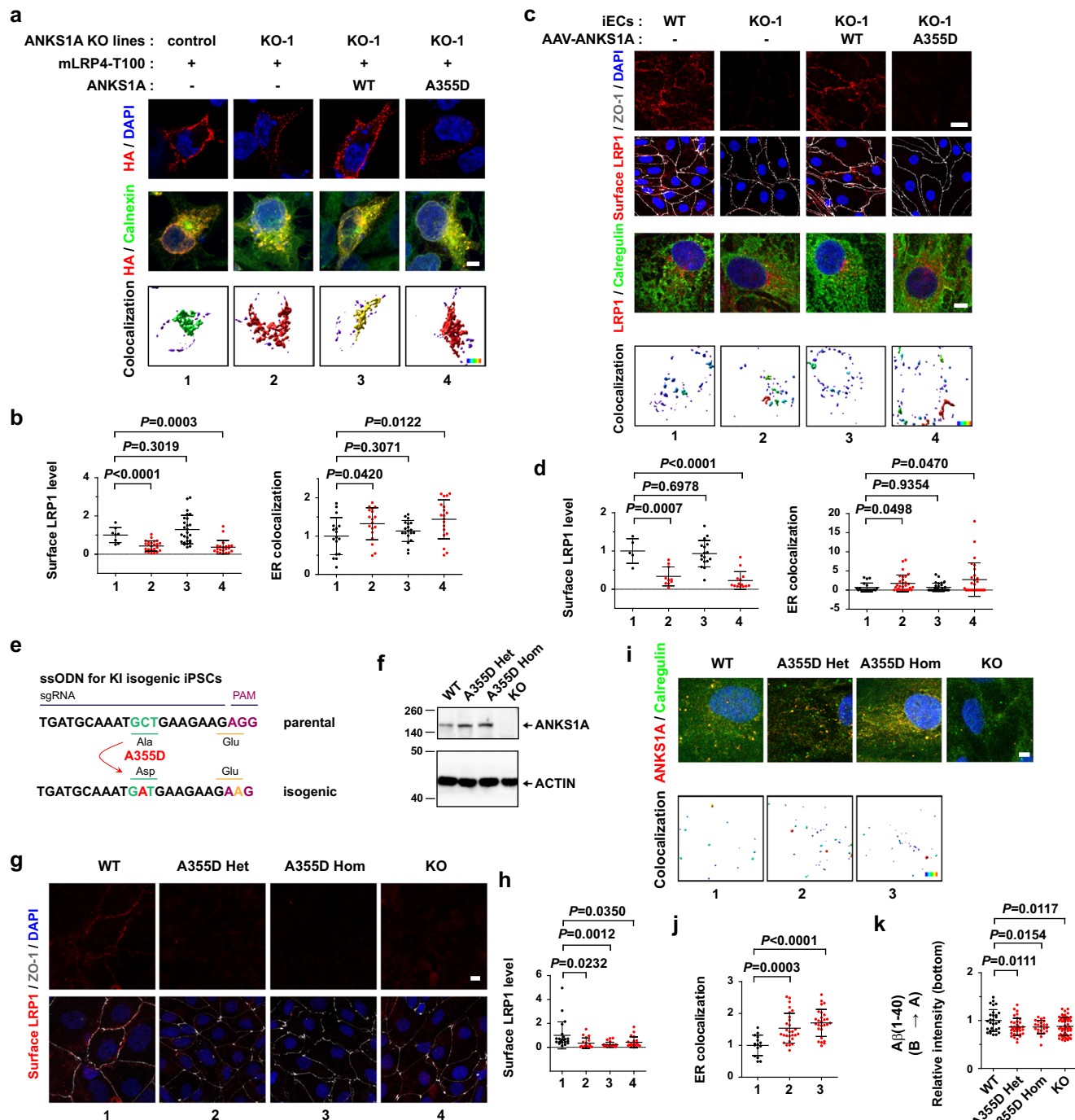

**Fig. 7 | *ANKS1A* A355D variant is defective in facilitating the ER export of LRP1.**
**a** Control or *ANKS1A*-deficient HEK293 cells were transfected with the LRP1 minireceptor expression vector with or without the *ANKS1A* expression vector. The surface LRP1 and ER-localized LRP1 levels were normalized to the control cells expressing the LRP1 minireceptor (scale bar, 5 μm). **b** *n* = 8, 24, 26, 21 cells for each group (left panel); *n* = 17, 18, 19, 19 for each group (right panel). **c** iECs were cultured and transduced with *AAV-BR1-A355D* virus. Surface LRP1 levels and ER-localized LRP1 were normalized to the WT iECs (scale bar, 20 μm (first panels), 5 μm (third panels)). The human iPSC-derived cells used in these experiments were generated from the GM23720 cell line. **d** *n* = 5, 10, 17, 14 for each group (left panel); *n* = 19, 30, 29, 28 for each group (right panel). **e** Schematic depicting the strategy for generating isogenic iPSCs expressing *ANKS1A* A355D. **f** Western blot analysis for expression of *ANKS1A* A355D in heterozygous (Het) or

homozygous (Hom) iECs. Results were confirmed at least in three independent experiments. **g, h** Experiments were performed as described in Fig. 7c, except that the iECs used were derived from the AG09173 cell line (scale bar, 10 μm). *n* = 22, 19, 26, 21 for each group. **i, j** Immunostaining was performed using ANKS1A-specific and ER-specific antibodies and localization to the ER was measured (scale bar, 5 μm). *n* = 15, 28, 27 for each group. **k** The human iPSC-derived cells were derived from the AG09173 cell line. *n* = 29, 36, 23, 54 independent wells for each group. The rainbow scale (**a, c, i**) represents the low (violet) to high (red) co-localization (colormap means statistics coded for intensity sum of fluorescences, $1.0 \times 10^4$ - $5.0 \times 10^7$ spectrum). Data (**b, d, h, j, k**) were from three independent experiments. Data in this figure are shown as mean ± SD. Two-tailed unpaired t-test against control or WT populations. Source data are provided as a Source Data file.

accumulation in the brain vessels and result in the pathogenic CAA commonly found in AD[5,13,15–21,23]. However, a study using LRP1-NPXY2 mutant (NPVYATL → AAVAATL) knock-in mice revealed lower Aβ deposition and plaques in the AD-like mouse model, suggesting LRP1's role in amyloidogenic processing of APP for Aβ clearance in neurons[38]. In such a mouse model with global LRP1 inactivation, a loss of LRP1 function favors the non-amyloidogenic APP processing, reducing the overall production of Aβ in the brain despite the impaired Aβ clearance. However, this impact of LRP1 on APP processing is barely present in the brain endothelial cells, where LRP1-mediated Aβ clearance plays a predominant role. It is interesting to note that the signals for APOE3 bound to non-vascular cells in the brain were lower in *ANKS1A* cKO mice than in control mice (see Fig. 2). It was found, however, that the levels of mature LRP1 in nonvascular tissues were not significantly different between the control and *ANKS1A* cKO groups, suggesting that their reduced binding to nonvascular cells was not the result of a reduced level of mature LRP1. It is known that more than 80 ligands are bound to LRP1 in various cell types[89,90]. Therefore, in endothelial *ANKS1A* cKO cases, ligands including APOE are likely to accumulate in the brain parenchyma due to inefficient LRP1-mediated cerebrovascular clearance. Indeed, we observed that the level of APOE in the nonvascular tissues was increased in endothelial *ANKS1A* cKO mice but not in control mice (see Supplementary Fig. 2l, m, third panels). This suggests that these ligands compete with exogenously added APOE3 for the LRP1 receptor in neurons and other brain parenchymal cells. This may explain the lower detection of APOE3 binding to the nonvascular cell types of *ANKS1A* cKO brain. Considering that *Tie-Cre; Ai9* mice showed that tdTomato was expressed exclusively in brain endothelial cells, the possibility of leaky Cre expression in other parenchymal cells is very low although we cannot completely rule it out. In agreement with our results, ANKS1A was barely detectable in non-vascular tissues of both control and *ANKS1A* cKO brains (see Supplementary Fig. 2l, m, second panels).

Evidences indicate that having reduced surface levels of brain endothelial LRP1 is a risk factor for CAA-associated AD pathogenesis[91]. Factors such as brain aging can also lead to a down-regulation of BBB transporters and receptors including LRP1, precipitating BBB breakdown with a gradual loss of proximal interactions among the principal cells of the BBB[18,19,31]. Our data demonstrate that certain necessary components from pericytes and/or astrocytes are crucial for maintaining the homeostatic levels of BBB proteins, and the levels of ANKS1A are highly susceptible to BBB disruption. Reduced levels of ANKS1A attenuate the cell surface levels of brain endothelial LRP1, causing inefficient cerebrovascular Aβ clearance. We hypothesized that down-regulation of both LRP1 and ANKS1A at the early stages of BBB breakdown can further impair the drainage of Aβ and aggravate deposition of insoluble Aβ on the vessel walls. Such a cycle change for the BBB is likely a critical precipitating factor for CAA pathogenesis. Our data revealed that ectopic expression of *ANKS1A* in the brain microvessels prevents the formation of insoluble Aβ peptides that result in neuronal death and cognitive behavioral impairments. It is noteworthy that our therapeutic approach did not reduce the levels of soluble Aβ peptides in the endothelium vessels. For the two groups of mice, namely *ANKS1A* KO vs. *AAV-ANKS1A* injected *ANKS1A* KO, both in the *5XFAD* background, the production of Aβ peptides was genetically unaffected. The vascular clearance of Aβ peptides in the injected group, however, was rescued, possibly affecting the extent of insoluble Aβ being formed in the vessels. It is possible that the formation of Aβ peptides in the initial fibrillogenesis for insoluble peptide are more effectively reduced in the injected group via LRP1-mediated cerebrovascular clearance. In addition, ANKS1A may also contribute to the parenchymal cell or glymphatic clearance pathways, which play a potential role in effective elimination of soluble Aβ peptides. These clearance pathways were not rescued by the expression of endothelial specific ANKS1A in *ANKS1A KO;5XFAD* mice, having a relatively large

amount of soluble Aβ peptides in the microvessels. It is possible that the types of soluble Aβ peptides in the microvessels of the injected group are totally different from those in the microvessels of the uninjected group. A future study will determine the validity of this hypothesis.

AAV vectors are currently considered to be the leading platforms for gene delivery for treatment of a number of human genetic diseases[92,93]. With two AAV-based therapeutics gaining recent approval by the US FDA and other candidates being tested in the clinic, AAV-mediated gene delivery may be an ideal means of targeted therapy[94–100]. With continued improvements in AAV-based technology, it is thought that AAV-vectors will lead to additional applications in the clinic, and these may include various means of addressing AD[101–104]. The recent advances in designing AAV capsids with a high degree of target specificity have further advanced the potential uses of AAV gene therapy. For example, the AAV-BR1 capsid variant displaying the NRGTEWD peptide demonstrates remarkable and unprecedented target specificity, having the strongest transgene expression in the brain capillary endothelial cells with little or no off-target expression such as the heart or liver[78]. For the mice model of incontinentia pigmenti, this selected viral capsid vector was very effective for decreasing the severe cerebrovascular defect of control mice after a single intravenous injection[105]. Likewise, the same vector expressing an LRP1 mini-gene restored the BBB integrity and protected the endothelial LRP1-deficient mice from neuronal loss and cognitive behavioral deficits[36,105]. Our data demonstrate that the cerebrovascular-specific expression of ANKS1A using the AAV-BR1 vector enhances the cell surface membrane localization of LRP1 and prevents neurotoxic and insoluble Aβ peptides from accumulating in the cerebrovascular tissues. Although we show that the viral expression of ANKS1A is sufficient to restore the LRP1-mediated Aβ clearance across the BBB in *ANKS1A* KO;*5XFAD* mice, the rescued phenotypes in these mice were not better than those of *5XFAD* mice. It remains to be determined whether the viral expression of ANKS1A does significantly improve the AD pathology in *5XFAD* mice and that they maintain a healthy BBB despite the persistent Aβ production. Anti-Aβ immunotherapy has been shown to reduce the amyloid burden in AD mouse models and also in the clinic for human trials[106–112]. However, as the amyloid levels in the endothelial vessels are also increased with immunotherapy, it is worth to test whether a combination of anti-Aβ immunotherapy with *AAV-ANKS1A* gene therapy would improve the effectiveness of such immunotherapy in AD-like animal models.

Human iPSC-derived cells have become indispensable in various research areas, improving reproducibility and use of custom human-origin isogenic cells for model building[113–115]. Using proven differentiation protocols[80], we developed a physiologically relevant iPSC-derived BBB model in a transwell format, consisting of a confluent monolayer of iPSC-derived brain endothelial cells on a permeable membrane with iPSC-derived pericytes and astrocytes layered on top. In this system, we found that the permeability of passively diffused soluble dextran was substantially lower than the permeability achieved in brain endothelial cell cultures with MEFs. This suggested that the reduced iBBB permeability was not an effect of physical layering of additional cells, but that it required the presence of certain cell types, namely both astrocytes and pericytes, thus validating the utility of the model. Our iBBB system was effective for testing Aβ transcytosis in an isogenic setting, and the experiments offered reproducible evidence for the role of ANKS1A in LRP1-mediated cerebrovascular Aβ clearance. This iBBB system also allowed us to demonstrate that the *rs6930932 ANKS1A* SNP variant manifests defective amyloid β clearance with its loss-of-function mutation. Given these findings, the iBBB system may facilitate additional studies for human CAA-associated AD and drug discovery.

In summary, our work has revealed a role for ANKS1A in the LRP1-mediated cerebrovascular clearance pathway; the functional

importance of ANKS1A is in maintaining a healthy BBB by facilitating clearance of insoluble amyloid fibrils in the endothelial vessels. As our study points to a role for ANKS1A in the pathophysiology of neuro-degeneration from vascular dysfunction, future studies for systemically examining the risk levels for the *rs6930932* variant and other *ANKS1A* variants in larger cohorts of AD patients could provide additional insights. In such studies, iBBB models from custom human iPSC-derived brain endothelial cells could enable assays for investigating factors affecting BBB function.

## Methods

### Ethics statement

Most of the research has been performed including local researchers from the Sookmyung Women's University, Daegu Gyeongbuk Institute of Science & Technology, and Ajou University School of Medicine. Roles and responsibilities were agreed upon amongst all the collaborators, and we have not discriminated against any individual based on gender, race, age, religion, sexual orientation, or disability status. The experiments on mice were conducted according to the institutional research guidelines of the Sookmyung Women's University and the protocol approved by the Institutional Animal Care and Use Committee (IACUC) of the university (protocol approval: SMWU-IACUC-2104-002-1).

### Mice

The *ANKS1A*[+/lacZ] gene trap and the *ANKS1A*[+/f] mice have been described[62,77]. Embryonic stem (ES) cell clones (CF0537 and EPD0578_4_D01) were purchased from the Mutant Mouse Regional Center and used to generate the *ANKS1A*[+/lacZ] and the *ANKS1A*[+/f] mice at the transgenic facility of Sookmyung Women's University. The *Tie2-Cre* mice (B6.Cg-Tg(Tek-cre)1Ywa) were obtained from RIKEN Institute (Japan)[69]. The *5XFAD* Tg (B6SJLTg(APPSwFlLon,PSEN1*M146L*L286V) 6799Vas/Mmjax) mice were obtained from The Jackson Laboratory (USA)[71]. *ANKS1B*[+/Lacz] mice were from Wellcome Trust Sanger Institute (United Kingdom)[116]. For studying aged mice, 18-month-old wild-type mice were purchased from Janvier Labs (France). For disrupting the BBB in mice, LPS dissolved in PBS was intraperitoneally injected into each mouse (10 mg/kg) and they were sacrificed after 24 h. The mice all have a C57BL/6 background. AAV injection experiments were conducted on 2-month-old mice. We examined AD pathology in 5 month-old *ANKS1A* KO mice and in 5-month- or 7-month-old *ANKS1A* cKO mice. Genotyping was performed using PCR analysis of tail genomic DNA and the primers pairs used for genotyping are provided in Supplementary Data 2. The Aβ pathology in *5XFAD* mice is higher in females than in males[73–76]. Thus, the rest of the Aβ pathology-related experiments were conducted on male mice to control for this gender difference; N = 5 for both *Tie2-Cre;5XFAD* and *ANKS1A*[f/f];*Tie2-Cre;5XFAD* mice or N = 3 for *ANKS1A*[+/f];*Tie2-Cre;5XFAD* mice in Fig. 3 and Supplementary Fig. 3; N = 5 for both *5XFAD* and *ANKS1A*[-/-]; *5XFAD* mice or N = 3 for *ANKS1A*[+/-]; *5XFAD* mice in Fig. 5 and Supplementary Fig. 5. In a separate experiment, we also performed Aβ pathological analysis for female mice; N = 3 mice for each group in Supplementary Fig. 5c–f. All mice had *ab libitum* access to standard laboratory diet and water. Sookmyung Women's University research guidelines and protocol approval: SMWU-IACUC-2104-002-1 were followed for the experiments on mice. All relevant animal experiments, including preparation for the culture, protein, and immunohistochemistry when applicable, used carbon dioxide ($CO_2$) inhalation as a method of euthanasia.

### Mouse brain primary cell culture

Mouse brain pericytes (PCs) and astrocytes (ASs) were prepared from neonatal mouse brains at P2[117,118]. The meninges were removed by rolling the brains on 3 M filter papers. Then, the brains were dissociated by passing 3–4 times through a 10 ml syringe with a 22-gauge needle. The cells were centrifuged (800 g, 7 min, 4 °C) and then the pellet was digested in low glucose DMEM (Sigma-Aldrich) containing collagenase/Dispase (100 mg/ml) and DNase I (4 µg/ml) for 10 min at RT. For PCs, the cells were centrifuged, resuspended in pericyte media (low glucose DMEM supplemented with 20% FBS, penicillin-streptomycin, heparin (750 U), ITS, and smooth muscle cell growth supplement), transferred into a T-75 flask and cultured for 1 week. For passing the mouse PCs, the cells were detached using 0.25% trypsin/EDTA when they reached 90 ~ 95% confluence. Following centrifugation, the cells were resuspended in fresh pericyte medium and allowed to settle in a flask for 1 h. Subsequently, the adherent cells were discarded and the cells in suspension were used for PC culture. For the ASs, the cells after tissue dissociation were centrifuged and resuspended in astrocyte media (low glucose DMEM supplemented with 20% FBS, penicillin–streptomycin and 15 mM HEPES). Mouse brain ASs were seeded on T-75 flasks coated with poly-L-ornithine and cultured for 10 days.

Primary brain endothelial cells were prepared from 1-month-old mice[117,119]. The meninges were removed by rolling the brains on filter papers. Then, the brains were dissociated by passing the tissue grinder and homogenized (10 brains in one tissue grinder). The homogenates were centrifuged (2580 g, 7 min, 4 °C) and the cell pellets were resuspended in 20% BSA and vortexed extensively. Afterwards, the precipitates were digested for 60 min at 37 °C in DMEM containing Collagenase/Dispase (100 mg/ml, Roche), DNase I (4 µg/ml, Roche), and TLCK (Nα-Tosyl-L-lysine chloromethyl ketone hydrochloride) (0.147 µg/ml, Sigma). After 60 min, the precipitates were centrifuged (2580 g, 7 min, 4 °C) and washed in PBS. Mouse brain endothelial cells were seeded on 24-well plates coated with type IV collagen. The brain endothelial cells were cultured in high glucose DMEM supplemented with 20% PDS (Plasma-derived bovine serum, FirstLink), penicillin/streptomycin, heparin (750 U), endothelial cell growth supplement (Sciencell) and 8 µg/ml puromycin (Sigma). After seeding, puromycin treatment was performed for two days, and subculturing was performed for 7 days.

### Human iPSC culture

The iPSC lines (GM23720 and AG09173) were purchased from the Coriell Institute for Medical Research. All human iPSCs were maintained in feeder-free conditions in mTeSR1 medium (Stem Cell Technologies) on Matrigel-coated plates (BD Biosciences). Brain endothelial cells, pericytes, and astrocytes were differentiated according to the published protocol[80]. For differentiation of brain endothelial cells (iECs), human iPSCs were dissociated with Accutase and reseeded onto Matrigel-coated six-well plates at 35,000 cells per cm² in mTeSR1 supplemented with 10 µM Y27632 (Stem Cell Technologies). Medium was replaced with fresh mTeSR1 daily. After 2 days, the medium was changed to DeSR1 medium (DMEM/F12 with Gluta-MAX (Life Technologies), supplemented with 0.1 mM β-mercaptoethanol, 1 X MEM-NEAA, 1 X penicillin-streptomycin and 6 µM CHIR99021 (R&D Systems)). Next day, the medium was changed to DeSR2 (DMEM/F12 with GlutaMAX (Life Technologies) supplemented with 1 X MEM-NEAA, 1 X penicillin-streptomycin and B-27 (Invitrogen)) and changed every day. After 5 days of DeSR2, the medium was changed to hECSR1 (Human Endothelial SFM (Thermo Fisher Scientific) supplemented with B-27, 10 µM retinoic acid (RA), and 20 ng ml⁻¹ basic fibroblast growth factor (bFGF)). Next day, the iECs were split using Accutase and reseeded with hECSR1 supplemented with 10 µM Y27632. The iECs were then cultured in hECSR2 medium (hECSR1 medium without RA and bFGF) and used within 1 week for experiments. For pericyte and astrocyte differentiation, we followed the published protocols[80].

### Generation of isogenic iPSCs with CRISPR/Cas9 genome editing

A guide RNA design tool from Integrated DNA Technologies (http://sg.idtdna.com) was used to design the sgRNAs. These included sgRNAs

for targeting the exon 3 of *ANKS1A* and generating a KO in isogenic iPSCs (forward 5′-CACCGAACGATGCGCTGACCAACG-3′, reverse: 5′-AA ACCGTTGGTCAGCGCATCGTTC-3′), and sgRNAs for targeting ANKS1A A355D sites for generating KI isogenic iPSCs (forward: 5′-CACCGT-GATGCAAATGCTGAAGAAG-3′, reverse: 5′-AAACCTTCTTCAGCATTTG CATCAC-3′). Single-stranded oligodeoxynucleotides (ssODN) were used to convert Ala (A) 355 in ANKS1A to Asp (D) 355 with a silent mutation at the protospacer adjacent motif (PAM) site to prevent recurrent Cas9-mediated cleavage in the edited cells (5′-CAGGTGA CGTGGAGAAAGCAGTGACTGAACTGATTATAGATTTTGATGCAAATG ATGAAGAAGAAGGTCCCTACGAAGCTCTGTATAATGCCATCTCCTGC CATTCGTTGGACAGCATGG-3′). The CRISPR/Cas9 genome editing was performed as described previously[120]. In brief, sgRNA was integrated into the CRISPR/Cas9 plasmid (pSpCas9(BB)−2A-GFP (PX458), Addgene #48138) and electroporated into iPSCs using the 4D-Nucleofector (Amaxa), and the P3 Primary Cell 4D Nucleofector kit (Lonza), following the manufacturer's instructions. After electroporation, the cells were resuspended with human ES (hES) media containing DMEM/F12, HEPES (Gibco) supplemented with 20% knock-out serum replacement (Gibco), NEAA (1X), GlutaMAX, 12 nM FGF2 (PeproTech, 100-18B), and 0.1 mM 2-mercaptoethanol (Sigma-Aldrich, M-3148) with 10 μM ROCK inhibitor (Tocris, 1254) and seeded on irradiated mouse embryonic fibroblasts (MEFs, Gibco, A34181). After the colonies were formed, each colony was transferred onto a well of a 12-well plate coated with MEFs and the culture was then maintained. In Supplementary Data 2, we provide the sequences of oligonucleotides used to confirm the generation of iPSCs as well as the off-target effects.

### In vitro BBB studies

The bEnd.3 mouse brain endothelial cells (ATCC® CRL-2299™) were cultured in DMEM supplemented with 10% fetal bovine serum in a 5% $CO_2$ incubator at 37 °C[66]. For gene knockdown experiments, 20 μM siRNA was electroporated into cells using the Neon Electroporation system, and the cells were then seeded onto a 2% Matrigel-coated Transwell polyester membrane cell culture insert ($1 \times 10^6$ cells cm$^{-2}$). All siRNA oligo duplexes were obtained from Origene, and the sequences are listed in Supplementary Data 2. At 24 h after seeding, mouse brain primary pericytes, astrocytes or moue embryonic fibroblasts (MEFs) were seeded on top of the bEnd.3 cells at a density of $5 \times 10^4$ cells per cm$^2$.

For iBBBs, iPSC-derived brain endothelial cells (iECs) were dissociated by Accutase, resuspended with hECSR1 supplemented with 10 μM Y27632 and then seeded onto a 24-well Matrigel-coated Transwell at a density of $10^6$ cells per cm$^2$ to complete a confluent monolayer for 24 h. Afterwards, the iPCs, iASs or MEFs were seeded on top of the iEC monolayer culture at a density of $5 \times 10^4$ cells per cm$^2$. The iBBBs were further cultured for 12 or 14 days.

For the paracellular permeability assays, FITC-labeled 70 kDa dextran (Sigma) and 4 kDa dextran (Sigma) were added at a final concentration of 2.5 nM. The Cy3-labeled Aβ (1−40) (Sigma) was added at a final concentration of 20 nM. For analysis of bottom (apical) to top (basolateral) permeability, 500 μl of medium containing the fluorescent proteins was added to the lower chamber; for analysis of top to bottom permeability, 100 μl medium of the fluorescent proteins was added to the top chamber. Permeability assays were conducted at 37 °C for 1 h. Media from the bottom chamber or the top transwell insert were collected and analyzed by a fluorescence plate reader (Molecular Devices). The rationale for using the indicated final concentrations for each protein was that the levels of diffused or transcytosed proteins were within a detectable range via fluorescence. To ensure that the added Aβ peptides did not disturb the integrity of our cellular BBBs, we also performed a paracellular permeability analysis following the transcytosis experiment. This involved first recovering the transcytosed Cy3-Aβ(1−40) for a fluorescent measurement, and

then the 4 kDa FITC-dextran being added to the same BBB setup at a final concentration of 2.5 nM for 1 h. With a comparative analysis of the diffused dextran, we confirmed that each BBB setup was not disrupted by the Aβ transcytosis assay.

### X-gal staining and immunostaining analysis

For X-gal staining, the brain sections were fixed with 1% paraformaldehyde (PFA), 0.2% glutaraldehyde, 1% deoxycholate, and 10% NP-40 in phosphate-buffered saline (PBS) for 10 min; the sections were then washed three times for 10 min and then incubated with the staining buffer ($K_4Fe(CN)_6$, $K_3Fe(CN)_6$, $MgCl_2$, X-gal) at 37 °C. After 15 h, the reaction was stopped by washing each slide three times with PBS.

HEK293 cells (ATCC® CRL-1573 ™) were cultured in high glucose DMEM (Sigma-Aldrich, D6429) supplemented with 10% FBS and 1% penicillin-streptomycin. Transient transfection was carried out using ViaFect (Promega, E4981), according to the manufacturer's instructions. The cells were plated at a density of $5.0 \times 10^5$ cells per 35-mm dish for immunostaining.

For cultured cells, they were fixed with 4% PFA, 2% sucrose in PBS for 15 min at room temperature. The cells were then washed with PBS and blocked in PBS containing 3% bovine serum albumin (BSA), 5% horse serum, and 0.5% Triton X-100 for 30 min. The cell samples were next incubated with primary antibodies diluted in 3% BSA in PBS at 4 °C. After an overnight incubation, the cells were washed, incubated with 3% BSA, 5% horse serum and 0.5% Triton X-100 in PBS for 30 min and treated with species-specific secondary antibodies at RT. After 2 h, the cell samples were washed with PBS and mounted with Prolong gold mounting medium (Invitrogen).

For the cell surface membrane staining of LRP1, the cultured cells were treated with anti-LRP1 rabbit IgG (Abcam) diluted in serum-free cultured medium at 4 °C. After an overnight incubation, the cells were washed, fixed, blocked and incubated with Alexa Fluor 647 conjugated ZO-1 antibody and Alexa Fluor 594 conjugated goat anti-rabbit IgG for 2 h. Then, the samples were washed and mounted with the mounting medium.

For the APOE3 binding analysis, vibratome sections were treated with recombinant APOE3 (PEPROTECH, final = 5 μg/ml) diluted in Hank's Balanced Salt Solution (HBSS, Gibco) at 4 °C. After an overnight incubation, the sections were washed, fixed and incubated with anti-APOE3 antibody. After 24 h incubation, the sections were washed, blocked, and incubated with FITC-conjugated IB-4 antibodies and TRITC-conjugated goat anti-rabbit IgG for 2 h. For the LRP1 antibody binding analysis, vibratome sections were treated with anti-LRP1 rabbit IgG antibody diluted in Hank's Balanced Salt Solution (HBSS, Gibco, final = 3 μg/ml) at 4 °C overnight. After the incubation, the sections were washed, fixed, and incubated with FITC-conjugated IB-4 antibodies and TRITC-conjugated goat anti-rabbit IgG for 2 h. The sections were then washed and mounted with the mounting medium. The same protocol was adopted for analyzing the internalization of Aβ peptides in brain microvessels, except for the incubation with Cy3-labeled peptides (final = 0.5 μg/ml) was carried out at 37 °C for 30 min.

For immunostaining analysis of the mouse brain cryosections, the sections were washed with PBS three times for 10 min at RT. Then, they were incubated in PBS containing 0.3% Triton X-100 for 30 min and treated with primary antibody diluted in PBS containing 0.3% Triton X-100 and 3% BSA for overnight at 4 °C. After the overnight incubation, the brain sections were washed, and treated with secondary antibodies for 2 h at RT. The sections were then three times washed with PBS containing 0.3% Triton X-100 and mounted with Prolong gold mounting medium (Invitrogen). Alexa Fluor 488-, 568, 647 conjugated anti-mouse, rabbit, rat, and goat antibodies were purchased from Invitrogen (Supplementary Data 3). A detailed description of all antibodies used is provided in Supplementary Data 3.

## GST pull-down assay, co-immunoprecipitation, and immunoblotting

Cultured cells or dissected mouse brains were lysed in PLC lysis buffer (50 mM HEPES pH 7.5, 150 mM NaCl, 10% glycerol, 1% Triton X-100, 1.5 mM MgCl$_2$, 1 mM EGTA, 10 mM NAPPi, 100 mM NaF) with protease inhibitor cocktail (Roche) on ice. Protein extracts were collected by centrifugation at 21,000 $g$ for 5 min. To perform the GST pull-down assay, LRP1-C (LRP1 cytoplasmic tail) GST fusion protein was purified with glutathione agarose (GE Healthcare). The GST- LRP1-C bound beads were then incubated with His-tagged PTB fusion protein of ANKS1A for 1 h. The beads were twice washed with the HNTG buffer (20 mM HEPES pH 7.5, 150 mM NaCl, 10% glycerol, 0.1% Triton X-100). The protein samples were separated by 10% sodium dodecyl sulfate-polyacrylamide gel electrophoresis (SDS-PAGE), and transferred onto a nitrocellulose membrane (Whatman). The membranes were blocked with 5% milk in TNTX (50 mM Tris-HCl, pH 7.6, 150 mM NaCl, 0.1% Triton X-100) for 30 min, then incubated with primary antibodies diluted in 3% BSA in TNTX, washed, and treated with anti-species specific secondary antibody conjugated with horseradish peroxidase (HRP) for 1 h.

For co-immunoprecipitation analysis, mouse primary brain endothelial cells were co-cultured with a mixture of astrocytes and pericytes in a transwell format for 3 days. Mouse primary brain endothelial cells were lysed in PLC lysis buffer and incubated with anti-LRP1 antibody (Abcam) for 90 min on ice while shaking occasionally. The Protein A-Sepharose 4B (Thermo Fisher Scientific) beads were added to cell lysates for 30 min at 4 °C. The beads were washed twice with the HNTG buffer and were separated by gradient SDS-PAGE (ATTO, Cat# 2331302).

The protein signals on the membranes were visualized and recorded with the ECL Plus Western Blotting Substrate reagent (Thermo Fisher Scientific) and a luminescent image analyzer (LAS 4000, Fuji Film), respectively. Protein band intensities were quantified using the ImageJ software.

## Preparation of crude ER membrane extracts

iECs were co-cultured iACs and iPCs in 3.5 cm transwell plates until 90–100% confluence was achieved to detect endogenous LRP1 expression. After 3 days, the cells were washed in 1X DPBS, treated with trypsin for 1 min at room temperature, and then harvested by centrifugation at 2000 $g$ for 5 min in B88-0 buffer (20 mM HEPES (pH 7.2), 250 mM sorbitol, 150 mM KOAc, and 5 mM MgOAc) containing 10 µg/ml soybean trypsin inhibitor. Cells were then permeabilized in ice-cold B88-0 with 40 µg/ml digitonin for 5 min and centrifuged at 9500 $g$ for 15 s to obtain crude ER membrane extracts. Solubilization buffer (1% SDS, 0.1% 2-mercaptoethanol, protease inhibitor cocktail (Roche)) was used to dissolve the extracts, which were then resolved using a gradient SDS-PAGE (ATTO).

## RNA analysis

Total RNA from the cultured cells or the brain microvessels was extracted using the TRIzol reagent (Invitrogen) and according to the manufacturer's instructions. For RT-PCR, cDNA was synthesized using approximately 4 µg of total RNA using a high-capacity RNA to cDNA kit (Applied Biosystems). Quantitative PCR was performed using a Light-Cycler 96 Instrument (Roche) with qPCRBIO SyGreen Blue Mix (PCR Biosystems). In the assays, GAPDH expression was used for normalization. The primers pairs used for qRT-PCR are listed in Supplementary Data 2.

For RNA sequencing, RNA purity was determined by assaying 1 µl of total RNA prep on a NanoDrop 8000 spectrophotometer. Total RNA integrity was checked using an Agilent Technologies 2100 Bioanalyzer with an RNA Integrity Number value. mRNA sequencing libraries were prepared according to the manufacturer's instructions (Illumina Tru-Seq stranded mRNA library prep kit). mRNA was purified and fragmented from 1 µg total RNA using poly-T oligo-attached magnetic beads and two rounds of purification. Cleaved RNA Fragments primed with random hexamers were reverse transcribed into first strand cDNA using reverse transcriptase, random primers, and dUTP in place of dTTP (incorporation of dUTP quenches the second strand during amplification, as the polymerase does not incorporate past this nucleotide). The cDNA fragments then had the addition of a single 'A' base and subsequent ligation of the adapter. The products were purified and enriched with PCR to create the final strand-specific cDNA library. The quality of the amplified libraries was verified by capillary electrophoresis (Bioanalyzer, Agilent). After the quantitative PCR (qPCR) using SYBR Green PCR Master Mix (Applied Biosystems), we combined the libraries that were index tagged by equimolar amounts in the pool. Cluster generation occurred in the flow cell on the cBot automated cluster generation system (Illumina). The flow cell was then loaded on a Novaseq 6000 sequencing system (Illumina), sequencing was performed with 2x100 bp read lengths. The differentially expressed genes based on RNA sequencing data are provided in Supplementary Table 1.

## Modified Y-maze test (spatial reference memory)

The modified Y-maze test was adapted from a previously published protocol[79]. The modified Y-maze apparatus consisted of three arms interconnected at an angle of 120°. The walls of the arms were 13-cm high and each was marked with a visual cue to serve as a spatial reference. For the test, first, with one arm of the maze closed, the test mice were allowed to explore the other two arms for 15 min before being returned to their home cage. One hour later, the mice were returned to the Y-maze and allowed to freely explore all three arms during a 5 min period while being video recorded. The time spent in the new arm and its frequency was quantified. The recorded videos were analyzed and tracked by the Smart 3.0 software.

## Vascular Aβ ELISA

Measurement of vascular Aβ was performed in accordance with the protocols[121,122]. In the published paper[121], the detailed protocol for the preparation of various buffer components is well explained in detail. The sample brains were dissected and their cerebral hemispheres were homogenized with a 10-ml syringe fitted with a 22-gauge needle three to four times in 10 ml PBS. Upon centrifugation, the homogenates were resuspended in 15% dextran solution and centrifuged again to remove myelin debris. The microvessel fragment pellet from each brain was weighed at this step and its weight (0.05–0.07 g per brain) was used to normalize the amyloid concentration measured by the ELISA. The brain microvessel fragments were resuspended in tissue homogenization buffer (2 mM Tris (pH 7.4), 250 mM sucrose, 0.5 mM EDTA, 0.5 mM EGTA, and protease inhibitor cocktail (Roche)) and stored at −80 °C after being divided into three aliquots ( ~150 µl for each tube). The aliquots were mixed with equal volumes (150 µl) of 0.4% DEA (diethylamine). Microvessel extracts were centrifuged at 135,000 $g$ for 1 h at 4 °C using an Optima LE-80K ultracentrifuge with a SW41 swinging bucket rotor (Beckman Coulter). The three DEA-containing supernatants (300 µl in each tube) from the same brain were mixed and neutralized with 90 µl of 0.5 M Tris-HCl (pH 6.8), and then assayed for Tris-base soluble Aβ levels in an ELISA (Invitrogen). A total of 200 µl of 95% formic acid (FA) was mixed with the remainder of the pellet in each tube and then sonicated by a Bioruptor in order to solubilize it. After centrifugation at 109,000 $g$ for 1 h at 4 °C, the supernatants were collected as FA-containing supernatants (~600 µl). The samples were then neutralized with 11.4 ml of 1 M Tris-base, divided into 1 ml aliquots, and an aliquot was used to measure FA-soluble Aβ levels using an ELISA (Invitrogen). A protocol for ELISA analysis was followed according to the instructions provided by the manufacturer. Amyloid concentrations were calculated using MyCurveFit software (https://www.mycurvefit.com/).

### In vivo administration and viral transduction with AAV vectors

The brain microvascular endothelial-specific AAV2-BR1 vector was described[78]. Mouse and human *ANKS1A* genes and *GFP* control were packaged into the AAV2-BR1 vector particles (Vectorbuilder). Mice at 2 months of age were put under anesthesia with isoflurane and the AAV viruses were injected through their tail vein at a dose of $2.5 \times 10^{11}$ GC (genome copies)/mouse. Behavior and tissue analyses were then performed when they reached 5 months of age.

For viral transduction experiments with iPSCs, the differentiated cells were incubated with AAV2 virus at a concentration of $1.0 \times 10^9$ GC/ml (Vectorbuilder).

### *ANKS1A* gene analysis of Alzheimer's disease (AD) patient samples

The genetic analysis via whole exome sequencing with the flanking intron has been described[123]. Briefly, the subjects for the genetic analysis were recruited consecutively from four university hospitals under the approval of their respective institutional review boards (IRBs) (AJIRB-BMR-SMP-18-545 and SCHBC_IRB_2012-124). All the subjects had passed both the clinical[124] and cerebrospinal fluid biomarker criteria for AD[125]. All participants and their caregivers (in cases of dementia) gave a written informed consent. The sex, number (*m:f* number) and age (mean ± standard deviation) of the participants were described in Supplementary Table 2; M:F = 12:27; 58.3 ± 5.0 year-old. The distribution of the *rs6930932* variant was comparable between male and female (Supplementary Data 1). The sex of the individuals in our country was determined by ID systems based on birth certificate by the doctors. This study is not a clinical trial. It is just an observation study without any manipulations. The sample sequencing was conducted using an Illumina HiSeq. 2500 instrument (mean coverage × 200.3, 100-bp) at Macrogen (Seoul, Korea). FASTQ-formatted read sequences were mapped to the reference human genome (hg19) using the Burrows-Wheeler Alignment Tool[126]. Duplicate reads were removed by applying the Picard tool (Picard Toolkit. Broad Institute, GitHub Repos. 2019. http://broadinstitute.github. io/picard/). Variant calling was performed using the Genome Analysis Toolkit (GATK)[127,128]. The variants were annotated using the SnpEff program[129].

### Image analysis

Image data were collected using an LSM700 (Carl Zeiss Microscopy) with a Plan-Apochromat ×20/0.8 M27 objective lens of an Axio Observer camera (Zeiss). The images were taken at 0.5–1 μm z-stack intervals over a 5–30 μm thickness. The fluorophore excitations were with 488, 555, and 639 nm laser wavelengths. All images were processed by the ZEN Black software.

The pericyte coverage was analyzed using the ZEN black software. Threshold of each fluorescence channel was adjusted with co-localized signals (yellow color) precisely selected on the scatterplot crosshair. This allowed the PDGFRβ-positive areas for the collagen (Col) IV-positive microvessels to be profiled in each microscopic field and to obtain Pearson's correlation coefficient.

Cerebral blood vessel network was analyzed using the AngioTool software[130]. For producing the vessel skeletonized images for Col IV staining, various parameters (e.g., vessel diameter) were optimized such that only the true vessels were labeled. The skeletonized images were further used to calculate length, total area, branching points and lacunarity of microvessels in each microscopic field.

Vascular binding of APOE3 and that of anti-LRP1 antibody was analyzed with the Imaris software, Bitplane 9.7.1. Likewise, internalization of the vascular bound Aβ40 was analyzed with the Imaris software. The specific signals bound to microvessels were rendered to 3D surfaces using the Imaris surface tool. The specific puncta within 0 μm from the IB4 or laminin-positive microvessels were sorted using the Imaris Filter feature and were defined as being vascular bound or being internalized.

Vascular amyloid accumulations, referred to as CAA, were analyzed with the Imaris software. The 6E10 antibody-positive amyloid beta and the Col IV-positive microvessels were rendered to 3D surfaces using the Imaris surface tool. The 6E10-positive puncta within 5 μm from the Col IV-positive microvessel were sorted using the Imaris Filter feature and defined as being a CAA. The number of CAAs was measured in each microscopic field.

Co-localization of LRP1 with the ER was closely analyzed with the Imaris software. The channel showing the ER in each cell was rendered to 3D surfaces and generating a masked channel. The new channel was then selected with the mask of the dataset (ROI, region of interest), and automatic thresholding was applied to the original confocal images. The "Build Coloc Channel" was selected for generating the colocalization channel. For generating a heatmap, the colocalization channel was further used to generate a new surface. The intensity sum (IS) of LRP1 within this surface was selected to generate the IS values in each cell. Maximal IS values were obtained from five independent cells in the group with the highest ER colocalization (e.g., *ANKS1A* KO cells) and the mean value applied to the cells in another group for generating its relative heatmap. For calculating the ER colocalization, the "% of ROI colocalized" among several choices in the colocalization channel was used. The mean ER colocalization value of the control cells was set to the value of 1.

NeuN-positive excitatory neurons were quantified using the Imaris software. In brief, NeuN-positive neurons and DAPI-positive nuclei in cortex and hippocampus were rendered to 3D surfaces using the Imaris surface tool. The DAPI positive nuclei overlapping with NeuN positive neurons were counted using the Imaris filter feature. Neurofilament SMI-312 was analyzed using the AngioTool software[130] as described in quantifying cerebrovascular network.

### Statistics

Statistical analysis was performed on data from three or four independent experimental replicates using GraphPad Prism 9.0. Error bars in the graphs represent the mean ± SD of the data. Statistical significance test was via the two-tailed unpaired *t* test for two samples in all the figures.

### Reporting summary

Further information on research design is available in the Nature Portfolio Reporting Summary linked to this article.

## Data availability

Source data are available for graphs plotted in Figs. 1–7 and Supplementary Figs. 1–7 and provided as a Source Data file. Scans of the full western blot gels can be found in the Source data file. All raw and analyzed sequencing data can be found at the NCBI Sequence Read Archive (accession number: GSE220105). All other data are available from the corresponding author upon request. Source data are provided with this paper.

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

## Acknowledgements

This work was supported by grants 2018M3C7A1056276, 2021R1A2C3011919, 2021R1A4A1027355 from NRF to S.P.; 2021R1C1C2009319 from NRF to H.L.; and HU23C0017 from KHIDI and KDRC to S.P.

## Author contributions

J.L., H.L., H.I.L., and M.S. designed and performed experiments and analyzed data. H.I.L. and J.S. generated and provided human isogenic

iPSCs. M.G.S. and E.L. analyzed the expression of ANKS1A from the published snRNaseq dataset. E.L. and S.A.P. analyzed the sequencing data of Korean EOAD patients. H.L. contributed to writing the manuscript. S.P. designed all experiments and wrote the paper.

## Competing interests

The authors declare no competing interests.
