## [Peer Review File · Nature Communications]

ANKS1A regulates LDL receptor-related protein 1 (LRP1)-mediated cerebrovascular clearance in brain endothelial cellsREVIEWER COMMENTS

Reviewer #1 (Remarks to the Author):

In the manuscript “A novel therapeutic target for enhancing cerebrovascular clearance by LDL receptor protein 1 (LRP1) in brain endothelial cells” by Jiyeon Lee and Haeryung Lee et al., the authors investigated the regulatory factor that determines the localization of LRP1 on cell surface. They focused on the effects of ANKS1A on LRP1 and demonstrated that endothelial deficiency of ANKS1A diminishes the cell surface LRP1 and reduces A β clearance across the BBB. They also found NPXY motifs of LRP1 was the key contributor for facilitating the transport of LRP1 from endoplasmic reticulum (ER) toward the cell surface. Furthermore, they used AD mouse models with ANKS1A deficiency and iPSC culture models mimicking ANKS1A deletion and proposed a therapeutic potential of ANKS1A by AAV-mediated overexpression. The manuscript is potentially of interest in that authors unveiled a novel function of ANKS1A as key factor for cell surface localization of LRP1. However, there are several major and a few minor concerns on the experimental methods, statistical analysis, and results in animal studies and iPSC-derived BBB models.

Major concerns/comments:

1. The number of mice used for analyses (N=3 per group) is too small. Also, it is not clear if both male and female mice were used or only one sex was used. It is well-recognized that the 5xFAD amyloid mouse model display strong sex-dependent neuropathology, thus a larger cohort size using both sex is critical.
2. The statistical analyses used is mentioned only for comparing two groups. Please provide the statistical analysis used for comparing groups of three or more.
3. The authors purchased two iPSC lines, but the data appears to be from one line. Please elaborate.
4. To ensure that ANKS1A^{f/f} alone has no impact on the mouse phenotypes, homozygous loxP (cre negative) control group needs to be added as part of the mouse model characterization.
5. The number of data points and the number of samples analyzed (taken from figure legends) do not match. For example, for Figure 2, the authors analyzed 10 microscopic fields per section for neocortex and 6 microscopic fields for hippocampus. However, there are more data points in the graphs. Same for the Figures 3 and 5.
6. The definitions of “vascular” and “nonvascular” binding of recombinant proteins (apoE and A β) and anti-LRP1 antibody in Figure 2 are not clear. Were they defined based on computer algorithms or by a study staff? If latter, steps to avoid bias needs to be described.
7. LRP1 is highly expressed in other brain cell types such as neuron, astrocyte, microglia, and OPC. In fact, endothelial LRP1 expression is relatively low compared to other cell types. In Figure 2c, there is hardly any signal in brain parenchyma which raise a question on staining and image acquisition protocols. Also, since the mouse model used here is endothelial cell specific knock-out of ANKS1A, nonvascular LRP1 signal should not be affected, but it is very difficult to assess from the data.
8. The authors demonstrated the effect of ANKS1A deletion on A β metabolism using transgenic mouse model of AD/CAA. They showed increase in vascular plaques in ANKS1A deletion mice in Figure 5. The parenchymal A β deposits need to be quantified with image analysis software to assess the changes in plaque load.
9. An AAV-mediated ANKS1A overexpression in ANKS1A knock-out mice led to a significant reduction in FA-soluble A β levels while Tris-soluble A β levels were not affected. The authors provided the potential underlying mechanism in lines 429-430. However, if the production of A β peptide is far

more dominant than the clearance, I would speculate that the A β levels in insoluble (FA-soluble) fraction would be either not changed or increased.

10. For AAV injection experiments, to ensure that the observed effect is not artifact of AAV injection itself, the control group needs to be injected with AAV-BR1-GFP. Also, the WT mice should be injected with AAV-ANKS1A and GFP.

11. AAV-mediated tissue specific expression of GFP was confirmed one month after the AAV injection, which does not match with the ANKS1A overexpression experiment that was assessed at three months after injection. The robust expression of transduced gene (GFP or ANKS1A) needs to be validated at the same time point.

12. The timeline of iPSC-derived endothelial cell (iECs) experiment is confusing. At which day of differentiation were the iECs plated on transwell, additions of the other cells (e.g., iPC, iAS, or MEF), TEER measurement, and BBB permeability measurements done? Perhaps, having a schematic drawing of experimental timeline in Figure 6 would be helpful. The TEER values derived from this differentiation protocol can widely fluctuate. The authors should demonstrate chronological changes of TEER values or indicate which day they measured TEER, with a clear rationale why they chose that day for TEER measurement.

13. In extended Figure 6, the authors validated purities of iPS cell-derived endothelial cells, pericytes, and astrocytes using bulk-RNA sequence. However, there is no statistical analysis done, and some of the genes, especially COI11A1, levels in endothelial cells were not upregulated compared with other cells. What was the rationale the gene list was chosen? Authors should also analyze more commonly studied genes like PDGFR β or CSPG4 (pericytes), and OCLN or CDH5 (endothelial cells).

14. The identification of ANKS1A p.A355D SNP in EOAD population is underpowered (only 39 EOAD cases). This finding needs to be replicated in a separate, larger cohort.

Minor concerns/comments:

1. Figure 1j. The title of the graph should be 4KDa Dextran bottom \rightarrow top not 4KDa Dextran top \rightarrow bottom.
2. The images for hippocampus are missing in Figure 2g.
3. Figure 2. Authors should state the concentration of each recombinant protein. It also needs rationale why those concentrations were chosen (from past reported literatures?)
4. Extended Figure 6a: there are 4 data points for each cell type, but the figure legend states that the data is from three independent bulk RNAseq. Which number is correct?

Reviewer #2 (Remarks to the Author):

General comments:

The manuscript "A novel therapeutic target for enhancing cerebrovascular clearance by LRP1" is in general clearly written and shows consistent studies from in vitro cell culture to mice – iBBB – population studies. All findings are shown on different levels (staining, ELISA, WB, mice, iPSC) and clearly presented. The authors show binding of ANKS1A to LRP1 validated using co-localization studies.

Major comments:

The authors measure the barrier integrity using dextran particles (Fig. 1), which is a valid method.

However, they apply 100µg/ml A beta peptides to the abluminal compartment and measure subsequently the amount of A beta at the luminal side. How can they be sure that this very high un-physiological amount of A beta peptides does not disturb the in vitro tightness of the barrier? It would be advisable to include a diffusion marker to the experiment and measure the transcytosis quotient!

The authors claim to show surface expression of LRP1, but rather show a more or less colocalization of LRP1 with ZO1. I would suggest performing a surface biotinylation experiment, to show increased surface expression of LRP1 and subsequent increased surface bound ligand internalization!

The abcam anti-LRP1 antibody, which is described as a specific antibody targeting binding domain II of LRP1, shows on the abcam website a distinct signal for approx. 90 kDa. As binding domain II of LRP1 is localized at the N-terminal region of LRP1 I would rather expect a corresponding band at approx. 515 kDa. There is either something wrong with the antibody or with the description in the manuscript.

The authors have not cited nor discussed the seminal work of the Roebrock laboratory on the NPXY deficiency in LRP1.

The authors describe the effect of ANKS1A on the LRP1 mini receptor. However, multiple studies of mutations and deletions in this region have been published before showing effects on LRP1 maturation, these have to be cited.

In figure 5 the authors use a conventional ANKS1A knock-out model and showed an increase in A beta plaque size. In 2019 Van Gool and colleagues showed an effect of NPXY alterations on A beta production in an mouse AD model and discussed this in regard to A beta clearance. How does this manuscript relate to the published data and why is this study not cited?

The AAV rescue experiment showed an interesting effect on insoluble A beta, but left soluble A beta alone. This is most peculiar, as many studies have demonstrated the opposite effect. This needs further clarification.

Lane 318: why are the authors using only a 70 kDa control dextran molecule, if they want to assess transport of a 4 kDa molecule (A beta). As mentioned above the authors did not control for potential toxic effects of A beta peptides which they apply in concentrations of 100µg/ml.

Fig 6 f does not properly show surface expression of LRP1. I suggest using biochemical methods to show an actual increase of surface LRP1 and subsequent functional internalization.

Most studies include Aβ1-40, what about Aβ1-42? There have been difference reported on the affinity of these peptides to LRP1.

I suggest a major revision.

Reviewer #3 (Remarks to the Author):

The manuscript by Lee et al is an interesting and extensive study on the role of ANKS1A in controlling LRP-1 function. In their manuscript the authors demonstrate that ANKS1A plays an important role in controlling cell surface expression of LRP-1 by binding to the intracellular NPXY motifs of LRP-1. As a consequence of this function, a deficiency in ANKS1A may lead to decreased trans-endothelial (and trans-BBB) transport – and its subsequent cerebrovascular accumulation - of the amyloid beta protein, which is a crucial process in the development of cerebral amyloid angiopathy. AAV-mediated delivery of ANKS1A in KO models rescued the effects of knock-out. Interestingly, a ANKS1A

variant encoding a loss-of-function protein was identified in the Korean population that was associated with reduced function of LRP-1.

Comments

Title and discussion: whereas the data are generally interesting and well-illustrated and supported by various sources of evidence, the step towards therapeutic application is probably too large at this point. The paragraph in the discussion starting at line 435 is therefore a bit out of scope and I would suggest (in the entire manuscript) to down-tune the claims and suggestions on the therapeutic potential of intervening with ANKS1A function, especially since some of the observed effects are relatively small (albeit significant).

Introduction: although it is tempting to cite (lines 56-57) those studies that show the highest numbers of association of CAA with AD, most accurate numbers are presented in a recent systematic review by Jakel et al (Alzheimers and dementia, 2021).

Methods:

- Endothelial cells, pericytes and astrocytes are characterized by expected levels of cell-specific markers. In this respect the endothelial expression of PDGF receptor beta seems unexpected. Can the authors explain this?
- What is the source of the bEND3 cells?
- The cellular BBB model: this is constructed by co-culturing pericytes, astrocytes (and fibroblasts as controls) on top of the endothelial cells for an extended period of time. Can the authors provide further evidence that the endothelial cells are not overgrown or replaced by any of these cell types after 12-14 days of cell culture?
- The time during which the permeability assays were done seems very short (only one hour); have the authors tested longer periods (e.g. 6-8 hours)?
- When assessing vascular amyloid beta levels (lines 690 etc.) there does not seem to be a control to test that the purification method indeed leads to isolation of microvessels. How has this been checked?
- The authors speak of 'vascular amyloid plaques' (e.g. line 749). This is a confusing terminology, since most scientists will refer to cerebrovascular amyloid beta accumulation as CAA, and not as plaques, which is typically used to describe parenchymal (non-vascular) depositions.
-

Other comments:

- PICALM seems to bind to the same epitope of LRP-1 as ANKS1A does; can the authors speculate if this leads to competition (with consequences for function)?
- There is probably a typing error in line 11, when referring to lane 13 (?).
- At some points the authors are overestimating the effects sizes of their observations. E.g. (lines 160-165): '...the ANKS1A-deficient Was ineffective in facilitating the transcytosis... '. The effect was only 10% and is thus not correctly described. Please check your manuscript for such firm claims that are not substantiated by the observations.
- In extended data 2e I do not see any (effect on) mouse IgG levels.
- In figure 2C LRP-1 expression is difficult to see.
- The staining for amyloid beta in figure a, does not look like the typical CAA as one would expect; in these pictures the amyloid accumulation seems to be more parenchymal than vascular.

Reviewer 1

Major comments

1. The number of mice used for analyses (N=3 per group) is too small. Also, it is not clear if both male and female mice were used or only one sex was used. It is well-recognized that the 5xFAD amyloid mouse model display strong sex-dependent neuropathology, thus a larger cohort size using both sex is critical.

From the suggestion, we have increased the number of mice used for experimental analyses in Fig. 2, Fig. 3 and Fig. 5. In *ANKS1A^{+/+};Tie2-Cre* vs. *ANKS1A^{ff};Tie2-Cre* group for Fig. 2 and Extended Data Fig. 2, we have used littermates for experiments. As sex differences have been reported in 5XFAD mice, including higher A β pathology in females than males, we focused on age-matched male mice for studying the role of ANKS1A in the 5XFAD brain (lines 220-223). We chose not to mix the genders for the experiments involving 5XFAD mice in Fig. 3 and Fig. 5. In addition, we studied the 5XFAD female group separately in our revised experiments and found that ANKS1A deficiency also aggravated the A β pathology of 5XFAD female mice similar to what we observed in 5XFAD males (lines 301-304).

2. The statistical analyses used is mentioned only for comparing two groups. Please provide the statistical analysis used for comparing groups of three or more.

As the reviewer suggests, we have included *ANKS1A^{ff}* mice in Fig. 2, *ANKS1A^{ff}; 5XFAD* mice in Fig. 3, and *ANKS1A^{+/-}; 5XFAD* mice in Fig. 5 for comparing the three groups of mice.

3. The authors purchased two iPSC lines, but the data appears to be from one line. Please elaborate.

A description of the iPSC lines (GM23720 and AG09173) is provided in the section on human iPSC culture in the methods. Specifically, the iPSCs used in Fig. 6a-i, Fig. 7c-d, Extended Data Fig. 6a-o, Extended Data Fig. 7b, and Extended Data Fig. 7f-h were derived from the GM23720 iPSC line, whereas those in Fig. 7e-k and Extended Data Fig. 7e were from the AG09173 line. Information regarding the origin of each iPSC-derived cell is added in the figure legends of the revised manuscript.

4. To ensure that ANKS1A^{ff} alone has no impact on the mouse phenotypes, homozygous loxP (cre negative) control group needs to be added as part of the mouse model characterization.

As the reviewer suggests, we have included ANKS1A^{ff} mice in Fig. 2 and ANKS1A^{ff}; 5XFAD mice in Fig. 3. Our results revealed that the homozygous loxP control group without Cre had no significant impact on the mouse phenotypes as shown in Fig. 2 and Fig. 3.

5. The number of data points and the number of samples analyzed (taken from figure legends) do not match. For example, for Figure 2, the authors analyzed 10 microscopic fields per section for neocortex and 6 microscopic fields for hippocampus. However, there are more data points in the graphs. Same for the Figures 3 and 5.

We have made corrections for the number of data points and the number of samples. Please see the legends for Fig. 2, Fig. 3, and Fig. 5 (lines 1241-1244; 1246-1249; 1259-1261; 1267-1268; 1301-1302; 1312-1316).

6. The definitions of “vascular” and “nonvascular” binding of recombinant proteins (apoE and A β) and anti-LRP1 antibody in Figure 2 are not clear. Were they defined based on computer algorithms or by a study staff? If latter, steps to avoid bias needs to be described.

The Imaris software, Bitplane 9.7.1, was used to analyze vascular binding of APOE3 and anti-LRP1 antibody. The Imaris software was also used to analyze the internalization of vascular bound A β 40 or A β 42. The specific signals bound to microvessels were rendered to 3D surfaces using the Imaris surface tool. Specific puncta within 0 μ m from the IB4-positive microvessels were sorted using the Imaris Filter feature and defined as vascular bound APOE3 or LRP1. For internalized vascular A β peptides, specific puncta within 5 μ m from the laminin-positive microvessels were sorted and defined as the internalized vascular A β peptides (image analysis). For normalization, a sum intensity of each puncta was divided by total area of IB4- or Laminin-positive microvessels in each microscopic field. An average intensity of each signal in WT control was set to 1 for calculating the relative intensity of each microscopic field (lines 835-840).

7. LRP1 is highly expressed in other brain cell types such as neuron, astrocyte, microglia, and OPC. In fact, endothelial LRP1 expression is relatively low compared to other cell types. In Figure 2c, there is hardly any signal in brain parenchyma which raise a question on staining and image acquisition protocols. Also, since the mouse model used here is endothelial cell specific knock-out of ANKS1A, nonvascular LRP1 signal should not be affected, but it is very difficult to assess from the data.

As shown in Fig. 2 and Extended Data Fig. 2, we observed that the signals for APOE3 and the anti-LRP1 antibody bound to cells in brain parenchyma were significantly lower in ANKS1A cKO than in WT mice. It is known that more than 80 different ligands are bound to LRP1. Therefore, in endothelial ANKS1A cKO, these ligands such as APOE are likely to accumulate in brain parenchyma due to inefficient LRP1-mediated cerebrovascular clearance. Our hypothesis is that these ligands would in turn compete with the exogenously added APOE3 or anti-LRP1 antibody for the LRP1 receptor expressed in neurons and other cell types of brain parenchyma. This hypothesis might explain why we observed the reduced level of APOE3 or anti-LRP1 antibody binding in the brain parenchyma of ANKS1A cKO mice (please see lines 211-216; 474-482).

8. The authors demonstrated the effect of ANKS1A deletion on A β metabolism using transgenic mouse model of AD/CAA. They showed increase in vascular plaques in ANKS1A deletion mice in Figure 5. The parenchymal A β deposits need to be quantified with image analysis software to assess the changes in plaque load.

As the reviewer suggested, we now show the extent of parenchymal A β deposits in Fig. 3 and Fig. 5 of the revised manuscript. In addition, we also show the levels of parenchymal A β deposits in Extended Data Figs 3 and 5.

9. An AAV-mediated ANKS1A overexpression in ANKS1A knock-out mice led to a significant reduction in FA-soluble A β levels while Tris-soluble A β levels were not affected. The authors provided the potential underlying mechanism in lines 429-430. However, if the production of A β peptide is far more dominant than the clearance, I would speculate that the A β levels in insoluble (FA-soluble) fraction would be either not changed or increased.

The reviewer points out that if the production of A β peptide is far more dominant than its clearance, the levels of A β in insoluble fraction would be either be not changed or be increased. We fully agree with this reasoning. In fact, we can reason that in the two groups of mice, ANKS1A KO;5XFAD vs. AAV-ANKS1A injected ANKS1A KO;5XFAD, the production of A β peptides would be basically unchanged. However, as the vascular clearance of A β peptides in the injected group would be improved by the viral expression of ANKS1A, this could alter the biochemical dynamics of insoluble A β formation in the vessels. We hypothesize

that the formation of A β peptides in the initial fibrillogenesis for insoluble peptides are more more efficiently eliminated via LRP1-mediated clearance pathway in the injected group than in the control, uninjected group. In addition, ANKS1A may also contribute to parenchymal cell or glymphatic clearance pathways, which play a potential role in effective elimination of soluble A β peptides. These pathways wouldn't be rescued by the endothelial specific ANKS1A expression in *ANKS1A KO; 5XFAD* mice, thus resulting in the higher levels of soluble A β peptides in the microvessels. Verification of the above points and their underlying hypotheses remain to be determined in a future research (lines 496-509).

10. For AAV injection experiments, to ensure that the observed effect is not artifact of AAV injection itself, the control group needs to be injected with AAV-BR1-GFP. Also, the WT mice should be injected with AAV-ANKS1A and GFP.

For the animal behavior experiment (Fig. 5d), we used a total of 8 *ANKS1A KO;5XFAD* mice as control, and all of them were injected with *AAV2-BR1-GFP* at 2 months of age. These mice were first used in the modified Y-maze task at 5 months of age, and then 3 of them were sacrificed for subsequent analysis of both their NeuN⁺ neurons and SMI312 neurites as shown in Fig. 5e and f. The other 5 mice were then added to the uninjected mice group for the A β analysis experiment using the ELISA shown in Fig. 5c. In addition, we also compared the *AAV2-BR1-ANKS1A* injected WT mice (N=4) with the uninjected WT mice (N=5) at 5 months of age. As shown in Extended Data Fig. 5j and k, we found that viral *ANKS1A* expression in WT mice did not affect the animal behavior and the resulting number of neurons compared with the uninjected WT mice. These results demonstrate that our AAV-related experimental results are not an artifact of AAV injection (lines 313-315; 335-336; 1312-1316; 1457-1461).

11. AAV-mediated tissue specific expression of GFP was confirmed one month after the AAV injection, which does not match with the ANKS1A overexpression experiment that was assessed at three months after injection. The robust expression of transduced gene (GFP or ANKS1A) needs to be validated at the same time point.

We also carried out the GFP expression analysis using the WT mice injected with AAV2-BR1-GFP virus at different time points. For example, at six months post AAV injection, we confirmed that the expression of GFP in WT mice remained strong enough for detection in the brain vessels (Extended Data Fig. 5j; lines 313-315; 1457-1461).

12. The timeline of iPSC-derived endothelial cell (iECs) experiment is confusing. At which day of differentiation were the iECs plated on transwell, additions of the other cells (e.g., iPC, iAS, or MEF), TEER measurement, and BBB permeability measurements done? Perhaps, having a schematic drawing of experimental timeline in Figure 6 would be helpful. The TEER values derived from this differentiation protocol can widely fluctuate. The authors should demonstrate chronological changes of TEER values or indicate which day they measured TEER, with a clear rationale why they chose that day for TEER measurement.

When we set up the iBBB experiments, we monitored for a chronological change of TEER values in WT iBBBs as shown in Extended Data Fig. 6c. Based on this data, we determined TEER measurement at 14 day after iBBB setup.

13. In extended Figure 6, the authors validated purities of iPS cell-derived endothelial cells, pericytes, and astrocytes using bulk-RNA sequence. However, there is no statistical analysis done, and some of the genes, especially COI11A1, levels in endothelial cells were not upregulated compared with other cells. What was the rationale the gene list was chosen? Authors should also analyze more commonly studied genes like PDGFR β or CSPG4 (pericytes), and OCLN or CDH5 (endothelial cells).

To address the issue of the statistical analysis for the iPSC-derived cells, we performed RT-qPCR analysis with the gene list shown in Extended Data Fig 6a. The differentially expressed gene list of each differentiated cell based on bulk-RNA sequencing was transferred to the supplementary Table 5. The gene list chosen for Extended Data Fig. 6a was based on the transcriptomic data analysis presented in the references (82-85) cited in our manuscript (lines 1078-1085). It seemed that PDGFR β was not strongly expressed in iPSC-derived pericytes. However, we were able to show that PDGFR β was strongly detectable in the co-culture conditions, suggesting that the proximal interactions among three cell types in BBB are crucial for modulating the protein level of PDGFR β (please see the reviewer only figure 1).

14. The identification of ANKS1A p.A355D SNP in EOAD population is underpowered (only 39 EOAD cases). This finding needs to be replicated in a separate, larger cohort.

We concur that it is essential to verify the frequency of the ANKS1A p.A355D SNP (rs6930932)

in a separate and larger cohort to confirm its relation to early-onset Alzheimer's disease as we have described it in the revised manuscript (lines 401-406). Unfortunately, we were not granted access to the exome sequencing database containing that for a large cohort of early-onset Alzheimer's disease (EOAD), such as the Alzheimer's Disease Sequencing Project (ADSP). Access to this database requires employment verification or contractual engagement with the National Institutes of Health (NIH), thereby precluding our ability to conduct a comprehensive analysis. Within the allotted time, we were able to calculate the frequency of the SNP rs6930932 by utilizing an easily accessible public database, the Genome Aggregation Database (gnomeAD). As gnomeAD represents a collection of population variations, it is only appropriate to compare the allele frequency of our cohort with that of the general population. In the gnomeAD database v.2.1.1, the ANKS1A p.A355D SNP exhibited a frequency of 0.008 and 0.005 among 3816 and 19952 allele numbers of the Korean and overall East Asian populations, respectively. We found that the frequency of *rs6930932* in our 39 samples was 5-10 times higher than in the East Asian populations in 1000 GP, gnomeAD, and the Korean in gnomeAD. We have included these findings in the revised manuscript with Supplementary Table 2 (see lines 556-561).

Minor comments

1. Figure 1j. The title of the graph should be 4KDa Dextran bottom → top not 4KDa Dextran top → bottom.

We have made the change to "4 kDa Dextran bottom → top" in Fig. 1j and then transferred the previous "4 kDa Dextran top → bottom" to Fig. 1l.

2. The images for hippocampus are missing in Figure 2g.

The Aβ42 peptide experiments are now shown in Fig. 2g and h.

3. Figure 2. Authors should state the concentration of each recombinant protein. It also needs rationale why those concentrations were chosen (from past reported literatures?)

We used 5 $\mu\text{g/ml}$ of APOE3 protein for each section, whereas we used 3 $\mu\text{g/ml}$ of anti-LRP1 antibody to analyze cell the surface-localized LRP1 levels. We also used 0.5 $\mu\text{g/ml}$ of A β peptides for each section in the internalization experiments (lines 697, 702 and 707). The concentrations were chosen based on trial and error as we determined each concentration to be appropriate for detecting the specific signals in each section. These experiments were also based on previously reported research (Ma et al., 2018, Molecular Neurodegeneration, 13:57).

4. Extended Figure 6a: there are 4 data points for each cell type, but the figure legend states that the data is from three independent bulk RNAseq. Which number is correct?

A total of 4 bulk RNAseq experiments were conducted (see supplementary Table 5).

Reviewer 2

1. The authors measure the barrier integrity using dextran particles (Fig. 1), which is a valid method. However, they apply 100µg/ml A beta peptides to the abluminal compartment and measure subsequently the amount of A beta at the luminal side. How can they be sure that this very high un-physiological amount of A beta peptides does not disturb the in vitro tightness of the barrier? It would be advisable to include a diffusion marker to the experiment and measure the transcytosis quotient!

There was a mistake in citing the final concentration of dextran and Aβ(1-40) used for the cellular BBB models. In fact, we used Cy3-labeled Aβ(1-40) at a final concentration of 20 nM in all the cellular BBB experiments. With this concentration in the abluminal side, the transcytosed Aβ(1-40) peptide was reproducibly measured with fluorescence detection. In addition, 20 nM Aβ(1-40) peptide has been used in other publications (Nature Medicine 2020, 26, 952). To ensure that the added Aβ peptides did not disturb the cellular BBB integrity, we always performed a paracellular permeability analysis following each transcytosis experiment; this included first recovering the transcytosed Cy3-Aβ(1-40) for a fluorescent measurement, and then the 4 kDa FITC-dextran was added to the same BBB setup at a final concentration of 2.5 nM for 1 hour. Through comparative analysis of the diffused dextran intensity, we confirmed that each BBB setup was not disrupted in the Aβ transcytosis experiment (lines 659-661; 666-674). Finally, our RAB11A/B knock-down experiments showed that Aβ peptides passed through the in vitro BBB systems via a RAB5-RAB11 dependent transcytotic pathway. These results indicate that our experimental condition for Aβ transcytosis did not disrupt the overall integrity of our in vitro BBB models (lines 167-169; 370-373).

2. The authors claim to show surface expression of LRP1, but rather show a more or less colocalization of LRP1 with ZO1. I would suggest performing a surface biotinylation experiment, to show increased surface expression of LRP1 and subsequent increased surface bound ligand internalization!

As the reviewer suggests, we performed both a cell surface binding and an internalization assay using HyLite488-labeled Aβ(1-40) peptide. As shown in Extended Data Fig. 1n, the Aβ(1-40) peptide was bound to the cell surface (no detergent treatment was performed) at 4 °C and more abundantly in control cells than in *ANKS1A* siRNA transfected cells. Likewise, the level of internalized Aβ(1-40) peptide after 15 min at 37°C was increased in control cells

than in *ANKS1A* siRNA transfected cells. These results are consistent with the surface level of LRP1 being prominently reduced in *ANKS1A* siRNA transfected cells (lines 150-153)

3. The abcam anti-LRP1 antibody, which is described as a specific antibody targeting binding domain II of LRP1, shows on the abcam website a distinct signal for approx. 90 kDa. As binding domain II of LRP1 is localized at the N-terminal region of LRP1 I would rather expect a corresponding band at approx. 515 kDa. There is either something wrong with the antibody or with the description in the manuscript.

We agree with the reviewer that the additional information on the anti-LRP1 antibodies need to be provided. For Fig. 2d and e, we used the anti-LRP1 antibody from R&D (Cat. No. AF4824), which is specific for the human LRP1 cluster III (Ser2522-Ile2941). For analyzing the cell surface levels of LRP1 in bEND.3 or iPSC-derived endothelial cells, we used the anti-LRP1 antibody from Abcam (Cat. No. ab92544), which was specifically raised against the 85 kDa subunit of LRP1. However, the exact epitope for this antibody is not known. Importantly, our experiments show that this antibody is very good at recognizing cell surface-localized LRP1 in intact cells (no detergent added to the cells) but not in *ANKS1A*-deficient cells, suggesting that this antibody likely recognizes the N-terminal extracellular portion of 85 kDa subunit of LRP1 (Extended Data Fig. 6n, o; lines 387-388).

4. The authors have not cited nor discussed the seminal work of the Roebrock laboratory on the NPXY deficiency in LRP1.

5. The authors describe the effect of *ANKS1A* on the LRP1 mini receptor. However, multiple studies of mutations and deletions in this region have been published before showing effects on LRP1 maturation, these have to be cited.

6. In figure 5 the authors use a conventional *ANKS1A* knock-out model and showed an increase in A beta plaque size. In 2019 Van Gool and colleges showed an effect of NPXY alterations on A beta production in an mouse AD model and discussed this in regard to A beta clearance. How does this manuscript relate to the published data and why is this study not cited?

We agree with the reviewer's comments that the additional papers showing the role of NPXY motifs in LRP1 should be cited. Please see the references cited in our manuscript (lines 83-87, 467-474).

7. The AAV rescue experiment showed an interesting effect on insoluble A beta, but left soluble A beta alone. This is most peculiar, as many studies have demonstrated the opposite effect. This needs further clarification.

As the reviewer suggests, we have included a hypothesis regarding the AAV-ANKS1A rescue effect on insoluble A β but not for the soluble A β in the Discussion part of our revised manuscript (lines 496-509).

8. Lane 318: why are the authors using only a 70 kDa control dextran molecule, if they want to assess transport of a 4 kDa molecule (A beta). As mentioned above the authors did not control for potential toxic effects of A beta peptides which they apply in concentrations of 100 μ g/ml.

As the reviewer points out, the 4 kDa dextran is a better control for A β transcytosis in iBBB experiments than the 70 kDa dextran. As we explained in Comment 1, we always performed a paracellular permeability analysis following a transcytosis experiment; we first assayed the transcytosed Cy3-A β (1-40) with a fluorescence measurement; then the 4 kDa FITC-dextran was added to the same BBB setup at a final concentration of 2.5 nM for 1 hour. Through comparative analysis of the diffused dextran intensity, we confirmed each BBB setup not being disrupted by the A β transcytosis assay (lines 659-661; 666-674). The permeability results with the 4 kDa dextran are added to the revised manuscript (see Fig. 6g).

9. Fig 6 f does not properly show surface expression of LRP1. I suggest using biochemical methods to show an actual increase of surface LRP1 and subsequent functional internalization.

As the reviewer suggests, we performed both cell surface binding and internalization assays using the HyLite488-labeled A β (1-40) peptide as we also addressed in Comment 2. The results are shown in Extended Data Fig. 6k (lines 381-383).

10. Most studies include A β 1-40, what about A β 1-42? There have been difference reported on the affinity of these peptides to LRP1.

We confirmed that in the iBBB experiments, A β (1-42) peptide was transcytosed in the WT iBBBs and this clearance was significantly reduced in ANKS1A KO iBBBs. We did not observe any differences between the A β (1-40) and A β (1-42) peptides in their transcytotic effects in the iBBBs; this is despite the affinity of A β (1-42) to LRP1 having been reported to be much lower than that of A β (1-40) (Deane et al., 2004, Neuron, 43, 333-344) (lines 377-378).

Reviewer 3

1. Title and discussion: whereas the data are generally interesting and well-illustrated and supported by various sources of evidence, the step towards therapeutic application is probably too large at this point. The paragraph in the discussion starting at line 435 is therefore a bit out of scope and I would suggest (in the entire manuscript) to downtune the claims and suggestions on the therapeutic potential of intervening with ANKS1A function, especially since some of the observed effects are relatively small (albeit significant).

We agree with the reviewer's comment that the step towards therapeutic application is too large at this point. We have changed the Discussion in the revised manuscript to accommodate the reviewer's comment (please see lines 527-537).

2. Introduction: although it is tempting to cite (lines 56-57) those studies that show the highest numbers of association of CAA with AD, most accurate numbers are presented in a recent systematic review by Jakel et al (Alzheimers and dementia, 2021).

As the reviewer suggests, we have cited the review paper to indicate the accurate number of the CAA-associated AD in our manuscript (please see lines 56-57).

3. Endothelial cells, pericytes and astrocytes are characterized by expected levels of cell-specific markers. In this respect the endothelial expression of PDGF receptor beta seems unexpected. Can the authors explain this?

We agree that the RT-qPCR data in Extended Data Fig. 1h were not properly explained. We explain that the relative level of ANKS1B was set to 1 in each cell type after normalization to the level of endogenous GAPDH in our previous manuscript. In the revised figure, the data were normalized by the level of GAPDH for each cell type. As shown, the relative level of PDGFR β was much lower in endothelial cells (a mean value, 0.007) than in pericytes (a mean value, 0.268).

4. What is the source of the bEND3 cells?

The designation for the bEND.3 cells in our manuscript is "bEND.3 (ATCC)" in the Methods section, which are endothelial cells isolated from brain tissue derived from a mouse with endothelioma. We corrected the designation in the revised manuscript text and figures (lines 646).

5. The cellular BBB model: this is constructed by co-culturing pericytes, astrocytes (and fibroblasts as controls) on top of the endothelial cells for an extended period of time. Can the authors provide further evidence that the endothelial cells are not overgrown or replaced by any of these cell types after 12-14 days of cell culture?

Per the comment, we examined the endothelial cells after 14 days of co-culture with pericytes and astrocytes. Our data revealed that the endothelial cells maintained a single monolayer and that the layer was not disturbed by pericytes and/or astrocytes (please see the reviewer only figure 2).

6. The time during which the permeability assays were done seems very short (only one hour); have the authors tested longer periods (e.g. 6-8 hours)?

We used Cy3-labeled A β (1-40) at a final concentration of 20 nM in all the cellular BBB experiments. With this concentration in the abluminal side for 1 hour, transcytosis of A β (1-40) peptide was reproducibly observed within a fluorescence detection range. To ensure that the added A β peptides do not disturb our cellular BBB integrity, we also always performed a paracellular permeability analysis following a transcytosis experiment; we first recovered the transcytosed Cy3-A β (1-40) for a fluorescence measurement, and then the 4 kDa FITC-dextran was added to the same BBB setup at a final concentration of 2.5 nM for 1 hour. Through a comparative analysis of the diffused dextran, we confirmed that each BBB setup was not disrupted by the A β transcytosis experiment. However, we found that the permeability of the 4 kDa dextran was significantly disrupted after A β (1-40) treatment when followed for more than 2 hours. After this optimization, we performed the assay for 1 hour for the A β (1-40) treatments in the cellular BBB experiments (lines 666-674).

7. When assessing vascular amyloid beta levels (lines 690 etc.) there does not seem to be a control to test that the purification method indeed leads to isolation of microvessels. How has this been checked?

We performed both immunostaining and Western blot analysis for the isolated microvessels to ensure that the microvessels were isolated with their integrity intact (Extended data Fig. 1g;

please see lines 125-126).

8. The authors speak of 'vascular amyloid plaques' (e.g. line 749). This is a confusing terminology, since most scientists will refer to cerebrovascular amyloid beta accumulation as CAA, and not as plaques, which is typically used to describe parenchymal (non-vascular) depositions.

From the suggestion, the term "vascular amyloid plaque" was replaced by "CAA" in the revised manuscript (please see lines 223-225; Supplementary Movie 1).

9. PICALM seems to bind to the same epitope of LRP-1 as ANKS1A does; can the authors speculate if this leads to competition (with consequences for function)?

PICALM is known to bind to the YXXP motif of LRP1 (Zhao, Z. et al. Nature Neuroscience 18, 978-987 (2015)), whereas ANKS1A does bind to the NPXY motifs of LRP1 as shown in Fig. 4.

10. There is probably a typing error in line 11, when referring to lane 13 (?).

We could not find this typing error.

11. At some points the authors are overestimating the effects sizes of their observations. E.g. (lines 160-165): '...the ANKS1A-deficient Was ineffective in facilitating the transcytosis... '. The effect was only 10% and is thus not correctly described. Please check your manuscript for such firm claims that are not substantiated by the observations.

From the suggestion, we made the correction in the statement, now stating: "However, compared to the control *in vitro* BBB, the ANKS1A-deficient BBB was not as effective in facilitating the transcytosis of A β (1-40) peptides" (please see lines 172-173).

12. In extended data 2e I do not see any (effect on) mouse IgG levels.

Mouse IgG-positive signals were marked as arrowheads, which can be seen at a higher magnification as shown in Extended Data Fig. 2e.

13. In figure 2C LRP-1 expression is difficult to see.

We modified Fig. 2c, for which, the LRP1 expression is better observed.

14. The staining for amyloid beta in figure a, does not look like the typical CAA as one would expect; in these pictures the amyloid accumulation seems to be more parenchymal than vascular.

For added clarification, we have included the supplementary Movie 1 where the amyloid deposits around the microvessels are shown.

Reviewer 4

1. Please provide a stronger evidence in support of the statement on line 92-92, “ANKS1A is specifically expressed in the endothelial cells of the brain”. For example, Betsholtz’s database [Single Cell RNA-seq Gene Expression Data (betsholtzlab.org)] shows that ANKS1A is also expressed in astrocytes, vascular smooth muscle cells, pericytes, microglia and oligodendrocytes.

As shown in Extended Data Fig. 1e, we performed X-gal staining analysis using brain sections to monitor for the specific ANKS1A expression, which were co-stained for markers of astrocytes, microglia and oligodendrocytes. We also analyzed the expression of ANKS1A in various cell types of hippocampus from the published snRNaseq dataset (GSE 166261) and had the highest expression of ANKS1A in brain endothelial cells (Extended Data Fig. 1f; Please see lines 121-125).

2. It would be beneficial to perform a co-immunoprecipitation/Western blotting experiment in primary brain vascular endothelial cells with and without silencing of ANKS1A to confirm the direct interaction of ANKS1A with endogenous LRP1. This data should be included in Figure 4.

As the reviewer suggests, we isolated the microvessels from mouse brains and then the cell lysates were incubated with anti-ANKS1A antibody for co-immunoprecipitation. We then used a gradient SDS-PAGE (ATTO, Cat# 2331302) to resolve the ER precursor form of LRP1, which is estimated to be approximately 504 kDa in size, based on having 4544 amino acids for mouse LRP1. As a result, we were able to detect the ER precursor form of LRP1 only in ANKS1A-precipitated protein complexes but not in control IgG-precipitated samples (Extended Data Fig. 4h; lines 282-283). The anti-LRP1 antibody used for the Western blot analysis was from Abcam (Cat. No. ab92544), which is specifically raised against the 85 kDa (β) subunit of LRP1. Although the exact epitope for this antibody is not known, our experiments show that this antibody was very good at recognizing the cell surface-localized LRP1 in intact cells (no detergent added to the cells), but not in ANKS1A-deficient cells, suggesting that this antibody likely recognizes the N-terminal extracellular portion of the 85 kDa subunit. This antibody, however, does not recognize the larger (α) subunit of cleaved LRP1 (Extended Data Fig. 6n, o; lines 387-388).

3. The receptor-associated protein (RAP) prevents premature binding of ligands to newly translated LRP1 in the ER, preventing aggregation and degradation of LRP1 instead of proper trafficking to the plasma membrane (Lillis et al., *Physiological Reviews* 2008; 88:887-918; PMID: 18626063). Do the authors know whether ANKS1A knockdown or affects the levels of RAP?

From the suggestion, we performed a Western blot analysis to see whether the levels of RAP were changed in ANKS1A deficient cells. In human iPSC-derived endothelial cells (iECs) lacking ANKS1A, the levels of RAP were similar to those of RAP in WT iECs (Extended Data Fig. 6g; lines 361-362).

4. In extended data Figure 8, the authors show that in the absence of functional ANKS1A protein after silencing, the surface levels of mature LRP1 are reduced, LRP1 is accumulated in ER fraction, and less LRP1 is shed as sLRP1. Could the authors corroborate their immunocytochemical findings by showing the accumulation of endogenous full-length LRP1 in ER fraction for example by Western blotting or using other independent biochemical methods after ANKS1A silencing?

For the suggestion, we used a digitonin (40 $\mu\text{g/ml}$) permeabilization to obtain crude ER extracts as described before (Lee et al., 2016, *Nature Communications*, 7, 12799) and were then subjected to Western blot analysis. As a result, we showed that the endogenous full-length LRP1 was more abundant in the ER fraction from ANKS1A KO iECs but not WT iECs. In this experiment, proteins were separated on a gradient SDS-PAGE gel (ATTO (Cat. No. 2331302)) and probed with an anti LRP1 antibody (Abcam (Cat. No. ab92544)) as described in the Comment 2 above (Extended Data Fig. 6l; lines 383-385).

5. Please provide details how ANKS1A^{+/+}; Tie2-Cre, ANKS1A^{f/f}; Tie2-Cre, ANKS1A^{+/+}; Tie2-Cre; 5XFAD, ANKS1A^{f/f}; Tie2-Cre; 5XFAD, ANKS1A^{+/+}; 5XFAD, and ANKS1A^{-/-}; 5XFAD mice were generated. Are all the animals on the same genetic background? Have the author's used littermate controls? What was the sex of the animals used and were there any sex differences? These details should be provided.

In ANKS1A^{+/+};Tie2-Cre vs. ANKS1A^{f/f};Tie2-Cre group for Fig. 2 and Extended Data Fig. 2, we used littermates for experiments. As sex differences have been reported in 5XFAD mice, including a higher A β pathology in females than males, we focused on age-matched male mice for studying the role of ANKS1A in the 5XFAD brain. We did not mix the genders for the experiments involving the 5XFAD mice in Fig. 3 and Fig. 5. We also studied the 5XFAD female

group separately in our revised experiments and found that ANKS1A deficiency also aggravated the A β pathology in the 5XFAD female mice similar to what we observed in the 5XFAD males (Extended Data Fig. 5c-f; lines 220-223; 299-304).

6. Is ANKS1A involved in the recycling of LRP1?

According to our co-localization study, GFP-tagged ANKS1A was also co-localized with RAB11, suggesting that ANKS1A may be involved in the recycling process (please see the reviewer only figure 3). However, the potential role of ANKS1A in the recycling of LRP1 does not explain why LRP1 is more accumulated in the ER of ANKS1A-deficient brain endothelial cells. It remains to be determined whether ANKS1A plays a role in the recycling of LRP1 in the brain endothelial cells.

7. Does ANKS1A knockout or silencing contributes to polarized localization of LRP1 to the basolateral side of endothelial cells? In Figure 1, can the authors demonstrate the polarized expression of LRP1 to the basolateral side of the b.END3 cells monolayer in control siRNA and ANKS1A siRNA treated cells?

As shown in Extended Data Fig. 1m, we observed that the polarized localization of LRP1 to the basolateral side of endothelial cells was not routed to the apical side. Instead, the cell surface level of LRP1 was decreased in ANKS1A siRNA-transfected cells (lines 148-150).

8. In Figure 2, can endothelial cell polarity and abundance of LRP1 in brain capillaries be shown in ANKS1A^{+/+}; Tie2-Cre, and ANKS1A^{f/f}; Tie2-Cre mice?

As shown in Extended Data Fig. 2k, the polarized expression of LRP1 to the abluminal side of endothelial cells was not misrouted to the luminal side by the ANKS1A loss. However, the surface expression of LRP1 was significantly reduced in ANKS1A-deficient endothelial cells (lines 201-203).

9. In Figures 3 and 5, can the authors provide supporting high-magnification images showing colocalization of endothelial cells with A β to support their vascular plaque classification?

As the reviewer suggests, we provide the supplementary Movie 1 showing that the vascular plaque (gray color) wrap or be in direct contact with the microvessels (red color).

10. Does endothelial and global knockout of ANKS1A affect parenchymal amyloid load?

For the suggestion, we now show the parenchymal A β deposits in Fig. 3 and Fig. 5 of the revised manuscript. In addition, we also show the parenchymal A β deposits in Extended Data Figs 3 and 5 (lines 238-240; 299-304).

11. In the data presented in Figure 5, it is unclear how the authors weighed isolated microvessels and determined TBS soluble and formic acid (FA) soluble A β 40 and A β 42 levels. How do the authors explain the levels of FA soluble A β 40 and A β 42 significantly less than in the TBS-soluble fraction in Fig 5c? Also, in the figure legend, the authors say N=3 mice per group but present 4-7 data points in panels 5c and 5d.

This part is explained in the Methods section of our revised manuscript. Briefly, after the brain homogenate was treated with 15% dextran to remove the myelin debris, the microvessel fragment pellet was weighed and the weight of the pellet was used to normalize the amyloid concentration measured in the ELISA. For the reviewer's suggestion, we present the number of mice used in the figure legends for Fig. 5c and 5d (lines 1312-1316).

12. Do the authors know for how long sustained elevation of ANKS1A persists in brain microvessels after AAV-BR1-ANKS1A injection in ANKS1A^{-/-}; 5XFAD mice (ANKS1A^{-/-}; 5XFAD + AAV-BR1-ANKS1A group) compared to ANKS1A^{-/-}; 5XFAD mice?

We also carried out the GFP expression analysis using the WT mice injected with AAV2-BR1-GFP virus at different time points. For example, six months after the AAV injection, we confirmed that the expression of GFP in WT mice remained strong enough to detect in brain vessels (Extended Data Fig. 5j; lines 313-315).

13. In Extended Data Figure 1, the authors show downregulation of RAGE expression in microvessels isolated from 22-month-old mice compared to 2-month-old mice by RT-qPCR. Earlier studies in rodents found that RAGE was upregulated significantly in brain microvessels with age (Osgood et al., *Neurobiology of Aging* 2017; 57:178-185; PMID:28654861). Please explain this discrepancy.

We found that RAGE expression was reproducibly down-regulated in microvessels from the 22 month old brains and this result was consistent with the published data (Yang et al., *Nature* 2020; 583: 425-430). We currently do not understand why our results are different from that by Osgood et al., as we also did not focus on the RAGE gene expression in our manuscript.

Redacted Reviewer Only Figures

REVIEWER COMMENTS

Reviewer #1 (Remarks to the Author):

The authors have adequately addressed my concerns. As such, the manuscript is somewhat improved.

Reviewer #2 (Remarks to the Author):

the authors answered all comments, therefore I suggest to accept the manuscript for publication.

Reviewer #3 (Remarks to the Author):

I have no further comments

Reviewer #4 (Remarks to the Author):

Nature Communications Manuscript Number: NCOMMS-22-45944A

Manuscript Title: "A novel therapeutic target for enhancing cerebrovascular clearance by LDL receptor related protein 1 (LRP1) in brain endothelial cells".

Authors: Lee et al.

Corresponding author: Dr. Soochul Park

Although the authors have addressed some of my concerns, they need to adequately address my earlier major concerns #2, #4, #9, and #11, listed below. Also, there are additional concerns regarding the new data presented by the authors. In new Figure 2, how do the authors explain a significant reduction of not only vascular but also non-vascular LRP1 as well in ANKS1Af/f; Tie2-Cre mice compared to ANKS1Af/f mice? This raises significant concern about the specificity of Cre expression in endothelial cells and, therefore, the data interpretation in 5XFAD; ANAS1Af/f and 5XFAD; ANAS1Af/f; Tie2-Cre mice presented in Figure 2.

2: It would be beneficial to perform a co-immunoprecipitation/Western blotting experiment in primary brain vascular endothelial cells with and without silencing of ANKS1A to confirm the direct interaction of ANKS1A with endogenous LRP1. This data should be included in Figure 4.

4: In extended data Figure 8, the authors show that in the absence of functional ANKS1A protein after silencing, the surface levels of mature LRP1 are reduced, LRP1 is accumulated in ER fraction, and less LRP1 is shed as sLRP1. Could the authors corroborate their immunocytochemical findings by showing the accumulation of endogenous full-length LRP1 in ER fraction, for example, by Western blotting or using other independent biochemical methods after ANKS1A silencing?

In new Extended data Figure 6I, the authors show the molecular size of the ER form of LRP1 around 460 kd. This contradicts the earlier work by Herz et al., EMBO J 1990; 9(6): 1769-1776, where the authors showed that proteolytic processing of the 600 kd LRP1 occurs in a trans-golgi compartment; therefore, the size of the uncleaved ER form of LRP1 should be ~600 kd. The authors need to provide methodological details and quantitative data.

9: In Figures 3 and 5, can the authors provide supporting high-magnification images showing the colocalization of endothelial cells with A β to support their vascular plaque classification?

The supporting data provided by the authors in response to #9 is not convincing. It shows that the microvessels are in the vicinity of amyloid plaques but not as you typically see in cerebral amyloid angiopathy (CAA). Therefore, it is also inappropriate to call it as CAAs in Figures 3b and 5b.

#11: In the data presented in Figure 5, it is unclear how the authors weighed isolated microvessels and determined TBS soluble and formic acid (FA) soluble A β 40 and A β 42 levels. How do the authors explain the levels of FA soluble A β 40 and A β 42 significantly less than in the TBS-soluble fraction in Fig 5c? Also, in the figure legend, the authors say N=3 mice per group but present 4-7 data points in panels 5c and 5d.

The authors need to provide a detailed methodology for others to reproduce their work by including information on the average weight of the microvessel pellet per mouse brain, the volumes of extraction buffers they used in each step, the final sample volume, and the dilution factor for ELISA assay. It is misleading to say Tris-soluble in Fig. 5c when the authors used dimethylamine (DEA) for extraction.

Reviewer 4 comments

Although the authors have addressed some of my concerns, they need to adequately address my earlier major concerns #2, #4, #9, and #11, listed below. Also, there are additional concerns regarding the new data presented by the authors. In new Figure 2, how do the authors explain a significant reduction of not only vascular but also non-vascular LRP1 as well in ANKS1A^{f/f}; Tie2-Cre mice compared to ANKS1A^{f/f} mice? This raises significant concern about the specificity of Cre expression in endothelial cells and, therefore, the data interpretation in 5XFAD; ANAS1A^{f/f} and 5XFAD; ANAS1A^{f/f}; Tie2-Cre mice presented in Figure 2.

2: It would be beneficial to perform a co-immunoprecipitation/Western blotting experiment in primary brain vascular endothelial cells with and without silencing of ANKS1A to confirm the direct interaction of ANKS1A with endogenous LRP1. This data should be included in Figure 4.

4: In extended data Figure 8, the authors show that in the absence of functional ANKS1A protein after silencing, the surface levels of mature LRP1 are reduced, LRP1 is accumulated in ER fraction, and less LRP1 is shed as sLRP1. Could the authors corroborate their immunocytochemical findings by showing the accumulation of endogenous full-length LRP1 in ER fraction, for example, by Western blotting or using other independent biochemical methods after ANKS1A silencing?

In new Extended data Figure 6I, the authors show the molecular size of the ER form of LRP1 around 460 kd. This contradicts the earlier work by Herz et al., EMBO J 1990; 9(6): 1769-1776, where the authors showed that proteolytic processing of the 600 kd LRP1 occurs in a trans-golgi compartment; therefore, the size of the uncleaved ER form of LRP1 should be ~600 kd. The authors need to provide methodological details and quantitative data.

9: In Figures 3 and 5, can the authors provide supporting high-magnification images showing the colocalization of endothelial cells with A β to support their vascular plaque classification?

The supporting data provided by the authors in response to #9 is not convincing. It shows that the microvessels are in the vicinity of amyloid plaques but not as you typically see in cerebral amyloid angiopathy (CAA). Therefore, it is also inappropriate to call it as CAAs in Figures 3b and 5b.

#11: In the data presented in Figure 5, it is unclear how the authors weighed isolated microvessels and determined TBS soluble and formic acid (FA) soluble A β 40 and A β 42 levels. How do the authors explain the levels of FA soluble A β 40 and A β 42 significantly less than in the TBS-soluble fraction in Fig 5c? Also, in the figure legend, the authors say N=3 mice per group but present 4-7 data points in panels 5c and 5d.

The authors need to provide a detailed methodology for others to reproduce their work by including information on the average weight of the microvessel pellet per mouse brain, the volumes of extraction buffers they used in each step, the final sample volume, and the dilution factor for ELISA assay. It is misleading to say Tris-soluble in Fig. 5c when the authors used dimethylamine (DEA) for extraction.

Responses to the reviewer's comments

Response to the comment 1: As shown in Fig. 2 and Extended Data Fig. 2, we observed that the signals for APOE3 and the anti-LRP1 antibody bound to cells in brain parenchyma were significantly lower in ANKS1A cKO than in WT mice. It is known that more than 80 different ligands are bound to LRP1. Therefore, in endothelial ANKS1A cKO, these ligands such as

APOE are likely to accumulate in brain parenchyma due to inefficient LRP1-mediated cerebrovascular clearance. Our hypothesis is that these ligands would in turn compete with the exogenously added APOE3 or anti-LRP1 antibody for the LRP1 receptor expressed in neurons and other cell types of brain parenchyma. This hypothesis might explain why we observed the reduced level of APOE3 or anti-LRP1 antibody binding in the brain parenchyma of ANKS1A cKO mice (please see lines 211-216; 474-482).

Response to comment 2: The reviewer suggested that we should confirm the direct interaction between ANKS1A and endogenous LRP1 using primary brain vascular cells. Two approaches were taken to address this issue. Firstly, microvessels from control brains were purified and used for preparing cell extracts, followed by the precipitation of endogenous LRP1 with control IgG or anti-LRP1 antibodies. We found that ANKS1a co-precipitates specifically with LRP1, but not with control IgG. Next, microvessels from WT or ANKS1A KO brains were purified for cell extracts and LRP1 was precipitated with anti-LRP1 antibodies. This experiment was repeated three times for quantification, and the level of ANKS1A was normalized with that of LRP1. In ANKS1A KO extracts, a nonspecific signal is observed at the same location as authentic ANKS1A in ANKS1A WT microvessel extracts. We observed, however, that the intensity of ANKS1A in WT microvessels was two times greater than the intensity of a nonspecific signal in KO microvessels. Based on these results, we conclude that ANKS1A and LRP1 have a specific interaction in brain microvessels.

Response to comment 3: The size of LRP1 is estimated to be 504 kDa, based on the fact that both mouse and human LRP1 have 4544 amino acids. A form of LRP1 of this size is unglycosylated, whereas the 600 kDa form of LRP1 mentioned by the reviewer is glycosylated. We used an anti-LRP1 antibody from Abcam (Cat. No. ab92544) to analyze crude ER extracts with Western blots. This antibody's specificity was demonstrated in Extended Data Fig. 6n; the antibody recognizes a 85 kDa subunit of LRP1, but not the larger (α) subunit of cleaved

LRP1. In addition, this antibody recognizes a discrete form of LRP1, which is likely to be an uncleaved form of LRP1. In iECs with or without Tunicamycin, we analyzed the MWs of glycosylated and unglycosylated forms of LRP1. Prestained protein markers were compared with immunoprecipitated LRP1 proteins. As shown in the reviewer only figure 1 at the end of this letter, LRP1's glycosylated form was approximately 530 kDa in size. In contrast, the unglycosylated form of LRP1 was approximately 463 kDa in size. Therefore, the endogenous LRP1 protein detected by co-immunoprecipitation followed by Western blotting was likely glycosylated. The small discrepancy in MWs in LRP1 between other groups and our lab may be due to experimental errors caused by the large size of the protein. In new Extended Data Fig. 6l, LRP1 levels were normalized with ribophorin levels, a protein marker specific to the ER. As a result, we showed that the endogenous full-length LRP1 was more abundant in the ER fraction from ANKS1A KO iECs but not WT iECs. An explanation of the detailed procedure for preparing the crude ER extracts using a digitonin (40 µg/ml) permeabilization was provided in the revised manuscript (please see lines 737-747). Please find it below for your convenience.

Preparation of crude ER membrane extracts

iECs were co-cultured iACs and iPCs in 3.5 cm transwell plates until 90–100% confluence was achieved to detect endogenous LRP1 expression. After 3 days, the cells were washed in 1X DPBS, treated with trypsin for 1 min at room temperature, and then harvested by centrifugation at 2000 rpm for 5 minutes in B88-0 buffer (20mM HEPES (pH 7.2), 250mM sorbitol, 150mM KOAc, and 5mM MgOAc) containing 10 µg/ml soybean trypsin inhibitor. Cells were then permeabilized in ice-cold B88-0 with 40 µg/ml digitonin for 5 minutes and centrifuged at 10,000 rpm for 15 seconds to obtain crude ER membrane extracts. Solubilization buffer (1% SDS, 0.1% 2-mercaptoethanol, protease inhibitor cocktail (Roche)) was used to dissolve the extracts, which were then resolved using a gradient SDS-PAGE (ATTO, Cat# 2331302).

Response to comment 4: CAA is a form of angiopathy in which amyloid beta peptide deposits in the walls of small to medium blood vessels in the brain and meninges. In our study, we focused on brain capillaries, where impaired LRP1-mediated A β clearance is linked to CAA. CAA pathology associated with brain capillaries is not a common type, however one of the reviewers suggested that cerebrovascular amyloid beta accumulation should be considered to be a form of CAA. The vascular A β plaques were analyzed and defined using the Imaris software in our first submission, but were renamed CAAs following a suggestion from reviewer #3. While there is a difference of opinion regarding terminology between two reviewers, we would like to keep this term in the revised manuscript. Our reviewer only figure 2 now shows more different types of A β deposits around the brain capillaries, along with a new supplementary video (please see the reviewer only figure 2 at the end of this letter).

Response to comment 5: It should be noted that DEA refers to diethylamine, not dimethylamine. In our new revised manuscript, this error has been corrected. Based on the reviewer's suggestion, the detailed protocol has been incorporated into the revised manuscript (see lines 787-809). The protocol is described below for your convenience.

Vascular A β ELISA

Measurement of vascular A β was performed as previously described^{121, 122}. In the published paper¹²¹, the detailed protocol for the preparation of various buffer components is well explained in detail. The sample brains were dissected and their cerebral hemispheres were homogenized with a 10-ml syringe fitted with a 22-gauge needle three to four times in 10 ml PBS. Upon centrifugation, the homogenates were resuspended in 15% dextran solution and centrifuged again to remove myelin debris. The microvessel fragment pellet from each brain was weighed at this step and its weight (0.05 ~ 0.07 g per brain) was used to normalize the amyloid concentration measured by the ELISA. The brain microvessel fragments were resuspended in tissue homogenization buffer (2 mM Tris (pH 7.4), 250 mM sucrose, 0.5 mM

EDTA, 0.5 mM EGTA, and protease inhibitor cocktail (Roche)) and stored at -80 °C after being divided into three aliquots (~150 μ l for each tube). The aliquots were mixed with equal volumes (150 μ l) of 0.4% DEA (diethylamine). Microvessel extracts were centrifuged at 135,000 g for 1 h at 4 °C using an Optima LE-80K ultracentrifuge with a SW41 swinging bucket rotor (Beckman Coulter). The three DEA-containing supernatants (300 μ l in each tube) from the same brain were mixed and neutralized with 90 μ l of 0.5 M Tris-HCl (pH 6.8), and then assayed for Tris-base soluble A β levels in an ELISA (Invitrogen). A total of 200 μ l of 95% formic acid (FA) was mixed with the remainder of the pellet in each tube and then sonicated by a Bioruptor in order to solubilize it. After centrifugation at 109,000 g for 1 hour at 4°C, the supernatants were collected as FA-containing supernatants (~600 μ l). The samples were then neutralized with 11.4 ml of 1 M Tris-base, divided into 1 ml aliquots, and an aliquot was used to measure FA-soluble A β levels using an ELISA (Invitrogen). A protocol for ELISA analysis was followed according to the instructions provided by the manufacturer. Amyloid concentrations were calculated using MyCurveFit software (<https://www.mycurvefit.com/>).

Redacted Reviewer Only Figures

REVIEWER COMMENTS

Reviewer #4 (Remarks to the Author):

We appreciate the authors' effort to address some of my concerns.

However, the authors have not provided an adequate explanation or data to address one of my major concerns (comment 1): "In new Figure 2, how do the authors explain a significant reduction of not only vascular but also non-vascular LRP1 as well in ANKS1Af/f; Tie2-Cre mice compared to ANKS1Af/f mice? This raises significant concern about the specificity of Cre expression in endothelial cells and, therefore, the data interpretation in 5XFAD; ANAS1Af/f and 5XFAD; ANAS1Af/f; Tie2-Cre mice presented in Figure 2." The author's response is speculative and without any experimental evidence. The data presented in Figure 2 implies that the Cre in ANKS1Af/f; Tie2-Cre mice is leaky (i.e., not endothelial-specific), as the authors show a significant reduction of both vascular and non-vascular LRP1. It is well known that LRP1 in other cell types in the CNS (For example, PMID: 15195085) plays a role in the uptake and degradation of A β . If LRP1 is reduced throughout the parenchyma, then that will also result in increased parenchymal and vascular A β deposition. Therefore, the author's conclusions and even the title of this manuscript are inconsistent with the presented data.

The authors have also not addressed my comment 2: "It would be beneficial to perform a co-immunoprecipitation/Western blotting experiment in primary brain vascular endothelial cells with and without silencing of ANKS1A to confirm the direct interaction of ANKS1A with endogenous LRP1. This data should be included in Figure 4." This data is critical to confirm the direct interaction of ANKS1A with endogenous LRP1 in primary brain vascular endothelial cells and the downregulation of LRP1 in the absence of ANKS1A and support the authors' hypothesis. The quality of blots and the quantification in Extended Data Fig. 4, panel h is not convincing, and the authors admit that there are nonspecific bands in Western blots performed with brain microvessels isolated from ANKS1A KO mice, raising concern about the specificity of ANKS1A antibody used in the study.

Reviewer 4 comments

We appreciate the authors' effort to address some of my concerns.

However, the authors have not provided an adequate explanation or data to address one of my major concerns (comment 1): "In new Figure 2, how do the authors explain a significant reduction of not only vascular but also non-vascular LRP1 as well in ANKS1A^{ff}; Tie2-Cre mice compared to ANKS1A^{ff} mice? This raises significant concern about the specificity of Cre expression in endothelial cells and, therefore, the data interpretation in 5XFAD; ANAS1A^{ff} and 5XFAD; ANAS1A^{ff}; Tie2-Cre mice presented in Figure 2." The author's response is speculative and without any experimental evidence. The data presented in Figure 2 implies that the Cre in ANKS1A^{ff}; Tie2-Cre mice is leaky (i.e., not endothelial-specific), as the authors show a significant reduction of both vascular and non-vascular LRP1. It is well known that LRP1 in other cell types in the CNS (For example, PMID: 15195085) plays a role in the uptake and degradation of A β . If LRP1 is reduced throughout the parenchyma, then that will also result in increased parenchymal and vascular A β deposition. Therefore, the author's conclusions and even the title of this manuscript are inconsistent with the presented data.

The authors have also not addressed my comment 2: "It would be beneficial to perform a co-immunoprecipitation/Western blotting experiment in primary brain vascular endothelial cells with and without silencing of ANKS1A to confirm the direct interaction of ANKS1A with endogenous LRP1. This data should be included in Figure 4." This data is critical to confirm the direct interaction of ANKS1A with endogenous LRP1 in primary brain vascular endothelial cells and the downregulation of LRP1 in the absence of ANKS1A and support the authors' hypothesis. The quality of blots and the quantification in Extended Data Fig. 4, panel h is not convincing, and the authors admit that there are nonspecific bands in Western blots performed with brain microvessels isolated from ANKS1A KO mice, raising concern about the specificity of ANKS1A antibody used in the study.

Responses to the reviewer's comments

Response to the comment 1:

The reviewer expresses concern that the Cre protein in ANKS1A^{ff}; Tie2-Cre mice may be leaky and expressed in nonvascular cell types, thus reducing the level of LRP on the surface of astrocytes, neurons and microglia. As a consequence, if this is the case, the reduction of non-vascular LRP1 would affect our conclusion and even our manuscript's title.

In response to the reviewer's concerns, we conducted two additional experiments. In the first step, we prepared microvessel and nonvascular cell extracts (described in Extended Data Fig. 1g) from Tie2-Cre (WT) and ANKS1A^{ff}; Tie2-Cre (cKO) mice for Western blot analysis. The anti-LRP1 antibody used in this experiment recognizes the extracellular region of the β -chain of mature LRP1 (described in Extended Data Fig. 6l, n). As shown in newly added Extended Data Fig. 2p and q, mature LRP1 levels were significantly reduced in the brain microvessels of cKO mice compared with control mice (left panel). However, the levels of LRP1 in

nonvascular tissues of cKO mice did not differ from those of control mice (right panel). We conclude from repeated experiments that Tie2-driven Cre expression does not alter the level of LRP1 in nonvascular cell types of ANKS1A cKO mice. The second step was to generate Tie2-Cre; Ai9 mice, which are able to exhibit robust tdTomato fluorescence following Cre-mediated recombination. The reporter mouse is the best model for testing the specificity of Cre expression in brain endothelial cells. As shown in Extended Data Fig. 2r, the whole brain of Tie2-Cre; Ai9 exhibits the exclusive expression of Cre in the brain vessels. Additionally, we performed immunostaining analysis using the brain endothelial markers, Col IV and ZO-1, to determine whether they co-localized with tdTomato fluorescence. Consequently, we were able to demonstrate that Cre driven by Tie2 is exclusively expressed in brain endothelial cells. Lastly, it appears that the term nonvas LRP1 in Fig. 2 and Extended Data Fig. 2 may be misleading since it may refer to the level of nonvascular LRP1. Thus, the name of nonvas LRP1 has been changed to α LRP1 Ab binding (nonvas). Together, our results indicate that ANKS1A cKO mice exhibit exclusive Cre expression in brain endothelial cells, but not in other parenchymal cells, supporting our conclusion that ANKS1A is crucial for LRP1 homeostasis in brain endothelium (please see lines 211-220; lines 477-489; lines 1436-1441).

Response to the comment 2:

A reviewer raises concerns about the specificity of our ANKS1A antibody.

Many experiments were conducted in this study to confirm the specificity of the ANKS1A antibody (please see Extended Data Fig. 1i, Extended Data Fig. 4b, Extended Data fig. 6e, and Fig. 7f). We agree with the reviewer that it is crucial to demonstrate that ANKS1A interacts directly with endogenous LRP1 in primary brain vascular endothelial cells, but the high noise/signal ratio made it technically difficult to demonstrate ANKS1A interaction with endogenous LRP1 in coimmunoprecipitation. To eliminate background in every immunoprecipitation, the extracts were precleared with control rabbit IgG. The

coimmunoprecipitation experiments were repeated six times and confirmed the interaction of ANKS1A with a full-length LRP1 in the microvessels of WT mice but not in those of KO mice (please see Fig. 4l, m). Furthermore, we found that ANKS1A cKO mice had a reduced level of mature LRP1 in their microvessels when compared to WT mice (please see Extended Data Fig. 2p, q). We hope our data will convince the reviewer that ANKS1A interacts with endogenous LRP1 in brain microvessels (please see lines 286-287; lines 1313-1319).

For your convenience, please find newly added figures below.

Reviewers' Comments:

Reviewer #4 (Remarks to the Author):

Nature Communications Manuscript Number: NCOMMS-22-45944D

Manuscript Title: "A novel therapeutic target for enhancing cerebrovascular clearance by LDL receptor related protein 1 (LRP1) in brain endothelial cells".

Authors: Lee et al.

Corresponding author: Dr. Soochul Park

We appreciate the author's effort in addressing my major concerns.

However, the authors still have not provided an adequate explanation or data to address one of my major concerns (comment 1): "In new Figure 2, how do the authors explain a significant reduction of not only vascular but also non-vascular LRP1 as well in ANKS1Af/f; Tie2-Cre mice compared to ANKS1Af/f mice? This raises significant concern about the specificity of Cre expression in endothelial cells and, therefore, the data interpretation in 5XFAD; ANAS1Af/f and 5XFAD; ANAS1Af/f; Tie2-Cre mice presented in Figure 2."

In the most recent revision, the authors make the point to convince us that non-vascular LRP1 levels are unchanged between the WT and ANSK1Af/f; Tie2-Cre mice, yet the quantification in figure 2d shows indicates a highly significant reduction in non-vascular LRP1 levels in ANSK1Af/f; Tie2-Cre mice, indicating gene leakage. This data casts doubts on endothelial-specific LRP1 deletion.

Next, the authors measured LRP1 levels in brain microvessels and microvessel-depleted (non-vascular cell extracts) from Tie2-Cre (WT) and ANSK1Af/f; Tie2-Cre (cKO) mice by western blot. Here, the authors used as a control mouse (Tie2-Cre) instead of ANSK1Af/f controls. These mice may or may not be on the same genetic background, and as shown in figure 2c, may express different levels of non-vascular LRP1.

The blots in figure 4l-m showing IP against LRP1 in microvessels are severely smeared and the ANKS1A band is not clearly visible. If the antibody is good, based on their in vitro data, why the ANSK1A band is not clearly visible? Why didn't the authors use primary brain endothelial cells as suggested previously in original comment 2?

Their explanation regarding ligand binding of LRP1 (lines 477-489) is speculative without any experimental evidence.

Reviewer 4 comments

We appreciate the author's effort in addressing my major concerns.

However, the authors still have not provided an adequate explanation or data to address one of my major concerns (comment 1): "In new Figure 2, how do the authors explain a significant reduction of not only vascular but also non-vascular LRP1 as well in ANKS1A^{ff}; Tie2-Cre mice compared to ANKS1A^{ff} mice? This raises significant concern about the specificity of Cre expression in endothelial cells and, therefore, the data interpretation in 5XFAD; ANAS1A^{ff} and 5XFAD; ANAS1A^{ff}; Tie2-Cre mice presented in Figure 2."

In the most recent revision, the authors make the point to convince us that non-vascular LRP1 levels are unchanged between the WT and ANSK1A^{ff}; Tie2-Cre mice, yet the quantification in figure 2d shows indicates a highly significant reduction in non-vascular LRP1 levels in ANSK1A^{ff}; Tie2-Cre mice, indicating gene leakage. This data casts doubts on endothelial-specific LRP1 deletion.

Next, the authors measured LRP1 levels in brain microvessels and microvessel-depleted (non-vascular cell extracts) from Tie2-Cre (WT) and ANSK1A^{ff}; Tie2-Cre (cKO) mice by western blot. Here, the authors used as a control mouse (Tie2-Cre) instead of ANSK1A^{ff} controls. These mice may or may not be on the same genetic background, and as shown in figure 2c, may express different levels of non-vascular LRP1.

The blots in figure 4l-m showing IP against LRP1 in microvessels are severely smeared and the ANKS1A band is not clearly visible. If the antibody is good, based on their in vitro data, why the ANSK1A band is not clearly visible? Why didn't the authors use primary brain endothelial cells as suggested previously in original comment 2?

Their explanation regarding ligand binding of LRP1 (lines 477-489) is speculative without any experimental evidence.

Responses to the reviewer's comments

We appreciate the reviewer's efforts to improve the quality of our work. In order to address the reviewer's concerns, we made the following efforts;

Comment 1:

The reviewer expresses concern that the Cre protein in ANKS1A^{ff}; Tie2-Cre mice may be leaky in nonvascular cell types, thus reducing the level of LRP on the surface of astrocytes, neurons and microglia. As a consequence, if this is the case, the reduction of non-vascular LRP1 would affect our conclusion. In response to the reviewer's concerns, we conducted additional experiments. First, we prepared microvessel and nonvascular cell extracts (as described in Extended Data Fig. 1g) from Tie2-Cre, ANKS1A^{ff} and ANKS1A^{ff}; Tie2-Cre (cKO) mice for Western blot analysis. The anti-LRP1 antibody used in this experiment recognizes

the extracellular region of the β -chain of mature LRP1 (described in Extended Data Fig. 6l, n). As shown in Extended Data Fig. 2p and q, mature LRP1 levels were significantly reduced in the brain microvessels of cKO mice compared with Tie2-Cre or ANKS1A^{ff} control mice (left and top panel). However, the levels of LRP1 in nonvascular tissues of cKO mice did not differ from those of Tie2-Cre or ANKS1A^{ff} control mice (right and first panel). We conclude from repeated experiments that Tie2-driven Cre expression does not alter the level of LRP1 in nonvascular cell types of ANKS1A cKO mice. Second, we observed that ANKS1A levels were barely detectable in the nonvascular tissues of cKO mice as well as those of Tie2-Cre or ANKS1A^{ff} control mice (right and second panel). As expected, however, the vascular ANKS1A was detectable in Tie2-Cre or ANKS1A^{ff} control brains but not in ANKS1A cKO brains (left and second panel). Third, we analyzed Tie2-Cre; Ai9 mice, which are able to exhibit robust tdTomato fluorescence following Cre-mediated recombination. The reporter mouse is the best model for testing the specificity of Cre expression in brain endothelial cells. As shown in Extended Data Fig. 2r, the whole brain of Tie2-Cre; Ai9 exhibits the exclusive expression of Cre in the brain vessels. In addition, we were able to demonstrate that Tie2-driven Cre is specifically expressed in brain endothelial cells using the brain endothelial markers, Col IV and ZO-1. Together, our results indicate that ANKS1A cKO mice exhibit a specific expression of Cre in brain endothelial cells and that the possibility of leaky Cre expression in other parenchymal cells is very low although we cannot completely rule it out (please see lines 211-223; lines 482-499; lines 1478-1483).

Comment 2:

The reviewer raises concerns regarding the specific interaction between ANKS1A and LRP1 in the brain endothelium.

We agree with the reviewer's claim that our coimmunoprecipitation experiments using brain microvessel extracts did not provide sufficient evidence to support the specific interaction

between ANKA1A and LRP1. As ANKS1A expression is dependent on interaction with pericytes and/or astrocytes, we had preferred using microvessel extracts in the previous experiments. Microvessel extracts had a high background, making it difficult to detect the ANKS1A protein associated with LRP1. In response to the reviewer's suggestion, we have used primary brain endothelial cells from WT and ANKS1A KO brain. With the help of conditioned medium derived from a mixture of pericytes and astrocytes, we have recently been able to obtain primary brain endothelial cells (please see Fig 4l-n). First of all, ANKS1A was prominently detected around the perinuclear ER in WT-derived primary brain endothelial cells, but not KO-derived primary brain endothelial cells (Fig. 4l). Furthermore, Western blot analysis of primary brain endothelial cell extracts confirmed the expression of both Occludin and Glut1 (glucose transporter 1), markers of brain endothelial cells (Fig. 4m, first panels). Finally, the cell lysates were incubated with the anti-LRP1 antibody for co-immunoprecipitation (Fig. 4m, second and third panels). The anti-LRP1 antibody was able to pull down the ER precursor form of LRP1 (approximately 530 kDa in size). The presence of ANKS1A was detected only in WT-, but not in ANKS1A KO-derived brain endothelial cell extracts (Fig. 4m, n). We hope our data will convince the reviewer that ANKS1A interacts with endogenous LRP1 in primary brain endothelial cells (please see lines 289-291; lines 615-628; lines 763-769; lines 1351-1360).

Comment 3:

The reviewer claims that our explanation regarding competitive ligand binding of LRP1 is speculative without any experimental evidence. In this revision, we provide experimental evidence that APOE, one of the major ligands for LRP1, was significantly increased in the nonvascular tissues of ANKS1A cKO brains in comparison with those of Tie2-Cre or ANKS1A^{fl/fl} control brains (Extended Data Fig 2p, q, third panels). Based on this result, we hypothesize that APOE but also other LRP1 ligands in ANKS1A cKO interfere with the binding of anti-LRP1 antibodies to LRP1 in nonvascular tissues and that this effect is due to inefficient LRP1-

mediated cerebrovascular clearance (please see lines 219-223; lines 489-495). The other LRP1 ligands that accumulate in the nonvascular tissues of ANKS1A cKO brains will need to be determined in our future work.

It is our hope that the reviewer will be satisfied with the explanation and data we have provided to address the major concerns. For your convenience, please find newly added figures below.

Extended Data Fig. 2

Fig. 4

REVIEWERS' COMMENTS

Reviewer #4 (Remarks to the Author):

In regards to point 1, we appreciate that the authors have now performed western blots to confirm that there is no LRP1 reduction in the parenchyma (non-vascular fraction) of ANKS1Af/f; Tie2-Cre mice. This seems in line with the author's hypothesis and claims throughout the manuscript. However, it is unclear why the authors continue to insist to present data in Figures 2c and d, indicating a reduction in parenchymal LRP1. This is inconsistent with the other data, and it is artifactual given the author's claims regarding the LRP1 antibody used. Moreover, it will cause confusion for the readers.

RESPONSE TO REVIEWERS:

We are grateful for the reviewer's agreement with our hypothesis and claims throughout the manuscript. As the reviewer suggested, we decided to eliminate the data in Figures 2c and d, as well as the supplementary Figures 2i and j from the previous manuscript. Additionally, the supplementary Figures 2g and h from the previous manuscript were moved to the newly revised Figures 2c and d. The text and figure legends were also revised to reflect this change. We hope the reviewer is satisfied with our modifications.